# Machine learning can identify newly diagnosed patients with CLL at high risk of infection

Rudi Agius[1,2], Christian Brieghel[2], Michael A. Andersen [2], Alexander T. Pearson [3], Bruno Ledergerber[4,5], Alessandro Cozzi-Lepri[6], Yoram Louzoun [7], Christen L. Andersen[2,8], Jacob Bergstedt [9], Jakob H. von Stemann[10], Mette Jørgensen [5], Man-Hung Eric Tang [5], Magnus Fontes[5,11], Jasmin Bahlo[12], Carmen D. Herling[12], Michael Hallek[12,13], Jens Lundgren [5], Cameron Ross MacPherson[5], Jan Larsen[1] & Carsten U. Niemann [2]*

Infections have become the major cause of morbidity and mortality among patients with chronic lymphocytic leukemia (CLL) due to immune dysfunction and cytotoxic CLL treatment. Yet, predictive models for infection are missing. In this work, we develop the CLL Treatment-Infection Model (CLL-TIM) that identifies patients at risk of infection or CLL treatment within 2 years of diagnosis as validated on both internal and external cohorts. CLL-TIM is an ensemble algorithm composed of 28 machine learning algorithms based on data from 4,149 patients with CLL. The model is capable of dealing with heterogeneous data, including the high rates of missing data to be expected in the real-world setting, with a precision of 72% and a recall of 75%. To address concerns regarding the use of complex machine learning algorithms in the clinic, for each patient with CLL, CLL-TIM provides explainable predictions through uncertainty estimates and personalized risk factors.

[1] Department of Mathematics and Computer Science, Technical University of Denmark, Lyngby, Denmark. [2] Department of Hematology, Rigshospitalet, Copenhagen University Hospital, Copenhagen, Denmark. [3] Department of Medicine, University of Chicago, Chicago, IL, USA. [4] University of Zurich, Zurich, Switzerland. [5] Centre of Excellence for Health, Immunity and Infections (CHIP), Rigshospitalet, Copenhagen University Hospital, Copenhagen, Denmark. [6] University College London, London, UK. [7] Department of Mathematics, Bar-Ilan University, Ramat Gan, Israel. [8] Department of Public Health, Copenhagen University, Copenhagen, Denmark. [9] Human Evolutionary Genetics Unit, Institut Pasteur, Paris, France. [10] Rigshospitalet, Copenhagen University Hospital, Copenhagen, Denmark. [11] International Group for Data Analysis, Institut Pasteur, Paris, France. [12] Department of Internal Medicine and Center of Integrated Oncology Cologne Bonn, University Hospital, Cologne, Germany. [13] Center of Integrated Oncology Cologne Bonn, University Hospital, Cologne, CECAD (Cluster of Excellence on Cellular Stress Responses in Aging-Associated Diseases), University of Cologne, Cologne, Germany. *email: Carsten.utoft.niemann@regionh.dk

Overall survival (OS) for patients diagnosed with chronic lymphocytic leukemia (CLL) has significantly improved with the introduction of combination chemotherapy, chemoimmunotherapy, and targeted therapy[1–4]. According to the international workshop on CLL guidelines, CLL treatment is not recommended unless cytopenia, symptomatic disease, or short lymphocyte doubling time is present;[5] most patients thus enter a "watch-and-wait" period at diagnosis. During this period, severe infections prior to CLL treatment result in a higher 30-day mortality (9.8%) than upon CLL treatment, leading to worse treatment-free survival and OS compared to matched patients without severe infection[6,7]. Predictive models for identifying these patients are warranted, since prognostic factors for infections in CLL prior to and upon CLL treatment are largely unknown. Resulting from the double-hit of immunosuppressive treatment (from chemoimmunotherapy) and immune dysfunction (from CLL)[2,3,8], infections have become the most common cause of death among all age groups in CLL[1]. Therefore, it is also necessary to model patients at risk of CLL treatment to identify those at risk of further immunosuppression. The first step in changing the natural history of CLL-induced immune dysfunction is thus to identify patients at risk of infection or CLL treatment at time of diagnosis[5]. Previously, models in CLL have focused on the prediction of progression — or treatment-free survival, and OS[9–11] — predictive models that combine treatment and infection as an outcome, such as we propose, have not yet been explored.

Fueled by the tumor microenvironment interaction, neoplastic CLL cells take hostage and impair normal B-cells, macrophages, and T-cell functions[12–16]. This causes hypogammaglobinemia, pseudo-exhausted immune function, and cytokine changes that may be partially reversed by targeted therapies[12,17–19]. If patients at high risk of infection or CLL treatment could be identified at time of diagnosis, targeted therapies modulating the immune dysfunction among these high-risk patients could be prospectively tested in a clinical trial. Thus, we developed the CLL Treatment-Infection Model (CLL-TIM), to select patients for a randomized clinical trial (PreVent-ACaLL, NCT03868722), investigating whether three months of venetoclax and acalabrutinib combination therapy can improve the natural history of immune dysfunction due to CLL. To our knowledge, this is the first time a machine learning model will be used for patient selection in a randomized clinical trial.

Current prognostic models in CLL[9–11,20–22], are based on a handful of variables extracted at the time of diagnosis or treatment, thus overlooking the complexity of CLL, and this, without being able to handle missing data or present uncertainty estimates. In order to address clinical viability, modeling of CLL-TIM was initiated at the International Modeling Immune System and Pathogen Camp[23] where collaborations between physicians, molecular biologists, and data analysts identified modeling requirements for trustable machine learning implementations in the clinic. Analogous to the different disciplines and experience of this team, we took a "multiple-outlooks" approach to the development of CLL-TIM, which was to collate an ensemble of 28 machine learning algorithms to model changes in patient histories spanning 7 years prior to CLL diagnosis. Histories of which included laboratory results, physician's decisions, infectious events and comorbidities along with traditional prognostic markers[11,24]. Validated on an internal Danish test cohort, and an independent external German cohort, CLL-TIM surpassed the current gold standard for prognostication in CLL (CLL-IPI[11]). As a result of our multiple-outlooks approach, we were able to substantially increase the number of patients detected as high-risk, and even under high rates of missing data, provided predictions for all CLL patients. By modeling both infectious disease and CLL treatment events as an outcome, we establish a link between immune dysfunction and progressive disease in CLL, and demonstrate the complexity and non-linearity of risk factors contributing to immune dysfunction and treatment need. Through our online version of CLL-TIM, CLL-TIM.org, we provide explainable predictions by accompanying them with uncertainty estimates and personalized risk factors driving a given patient's predicted risk.

## Results

**Patient characteristics from Danish National CLL registry.** Based on the Danish National CLL registry, we identified 4149 patients diagnosed with CLL between January 2004 and July 2017. To ensure inclusion of all results from tests ordered at time of diagnosis we shifted time-point zero, which we refer to as the prediction point, to three months post-diagnosis for the training phase. We thus excluded patients who died ($n = 74$) or initiated CLL treatment ($n = 373$) prior to this (Supplementary Fig. 1), reducing the available sample size to $n = 3720$ (see Table 1 for baseline characteristics). For assessment in the internal test cohort, time-point zero varied between zero and three months post-diagnosis and for the external test cohort it varied between zero and 1-year post-diagnosis (Supplementary Fig. 2). For modeling, we only used patient data prior to the prediction point. As a composite outcome, we set out to predict the combined event of an infection or CLL treatment within 2-years from the prediction point. As it is standard practice that a blood culture is drawn when a patient has symptoms classified as a serious clinical infection, we used the event of having a blood culture drawn as a proxy for infection. This, irrespective of the result of blood culture being negative, indicative of contamination or positive[25–27]. As a first event during the 2-year predictive window, 572 (15.4%) patients had a severe infection, 398 (10.7%) received CLL treatment and 103 (2.8%) died, while 2647 (71.1%) had no study-relevant events (Supplementary Fig. 1). All CLL-IPI variables[11] were available for 48% of the cohort. Using stratified sampling that preserved class distributions, we randomly divided the cohort into a training set of 2432 (65%) patients and equally sized internal validation and test sets (~17.5% each at 642 and 646 patients, respectively).

**Development and Composition of CLL-TIM.** For each patient, we used three look-back windows of 3 months, 1 year, and 7 years prior to CLL-diagnosis to model microbiology, laboratory, pathology, clinical and CLL-specific patient data (Fig. 1a–c; Supplementary Methods subsection Feature Generation). Within these windows we used features like the Bag-Of-Words[28] (BOW), which describes the frequency of past events. Other features were designed to capture: the density and recentness of infections (Supplementary Fig. 3); rates of change; variability; and minima and maxima of laboratory test results, among others (Supplementary Table 1). We further modeled information related to the date of routine laboratory tests to capture the urgency of a patient's condition and symptomology as interpreted by the physician (Supplementary Methods subsection Feature Generation). This resulted in a final feature space of 7,288 dimensions (Supplementary Table 2), reduced using dimensionality reduction techniques (Fig. 1d, Supplementary Table 3 and Methods subsection Base-learner generation), upon which we applied 2,000 different algorithms (referred to as base-learners) — each providing a unique outlook into the patient's history (Fig. 1d; Methods subsection Base-learner generation). We next generated 29 ensembles (of sizes 2–30 base-learners) using a genetic algorithm (Fig. 1e; Methods subsection Ensemble generation), ranked the 29 ensembles using an ensemble diversity and generalization score (Methods subsection Ensemble ranking); from which the

**Table 1 Baseline characteristics internal and external cohorts.**

| Variable | Level | Internal Train (n = 2432) | Internal Validation (n = 642) | Internal Test (n = 646) | Internal Total (n = 3720) | External CLL7 (n = 365) |
|---|---|---|---|---|---|---|
| Age (years) | <65 years | 708 (29.1) | 218 (34.0) | 188 (29.1) | 1114 (29.9) | 269 (73.7) |
| | ≥65 years | 1724 (70.9) | 424 (66.0) | 458 (70.9) | 2606 (70.1) | 96 (26.3) |
| Sex | Female | 955 (39.3) | 274 (42.7) | 251 (38.9) | 1480 (39.8) | 131 (35.9) |
| | Male | 1477 (60.7) | 368 (57.3) | 395 (61.1) | 2240 (60.2) | 234 (64.1) |
| Binet stage | A | 2068 (85.0) | 558 (86.9) | 551 (85.3) | 3177 (85.4) | 365 (100) |
| | B | 303 (12.5) | 63 (9.8) | 77 (11.9) | 443 (11.9) | 0 (0) |
| | C | 61 (2.5) | 21 (3.3) | 18 (2.8) | 100 (2.7) | 0 (0) |
| β2 microglobulin > 4 mg L$^{-1}$ | No | 1620 (87.6) | 434 (90.0) | 444 (87.2) | 2498 (88.0) | 345 (94.5) |
| | Yes | 229 (12.4) | 48 (10.0) | 65 (12.8) | 342 (12.0) | 4 (1.1) |
| | missing | 583 | 160 | 137 | 880 | 16 (4.4) |
| IgHV status | mutated | 1320 (69.3) | 373 (72.9) | 370 (71.2) | 2063 (70.3) | 275 (75.3) |
| | unmutated | 584 (30.7) | 139 (27.1) | 150 (28.8) | 873 (29.7) | 85 (23.3) |
| | missing | 528 | 130 | 126 | 784 | 5 (1.4) |
| Hierarchical FISH[a] | del(17p) | 108 (4.4) | 25 (3.9) | 25 (3.9) | 158 (4.2) | 9 (2.5) |
| | del(11q)[b] | 130 (5.3) | 31 (4.8) | 36 (5.6) | 197 (5.3) | 28 (7.7) |
| | Trisomy12[c] | 244 (10) | 61 (9.5) | 73 (11.3) | 378 (10.2) | 27 (7.4) |
| | Normal[d] | 440 (18.1) | 127 (19.8) | 141 (21.8) | 708 (19.0) | 298 (81.6) |
| | del(13q)[e] | 981 (40.3) | 251 (39.1) | 222 (34.4) | 1454 (39.1) | – |
| ECOG performance status | 0 | 1859 (76.4) | 502 (78.2) | 515 (79.7) | 2876 (77.3) | 312 (85.5) |
| | 1 | 459 (18.9) | 112 (17.4) | 100 (15.5) | 671 (18) | 48 (13.2) |
| | 2 | 71 (2.9) | 20 (3.1) | 19 (2.9) | 110 (3) | 1 (0.3) |
| | 3 | 25 (1) | 4 (0.6) | 9 (1.4) | 38 (1) | 0 (0) |
| | 4 | 7 (0.3) | 1 (0.2) | 1 (0.2) | 9 (0.2) | 0 (0) |
| | missing | 11 (0.5) | 3 (0.5) | 2 (0.3) | 16 (0.4) | 4 (1.1) |

Baseline characteristics for n = 3720 patients after excluding patients that initiated CLL treatment or died before the prediction point of 3-months post-diagnosis (Supplementary Fig. 1). As expected for a population-based cohort of patients at time of CLL diagnosis, 70% were above 65 years of age, 60% were male, 30% had IgHV unmutated status, 12% had elevated β2 microglobulin, and 15% were Binet stage B or C. German External CLL7 cohort had 26% of patients above 65 years of age, 64% were male, 23% had IGHV unmutated status, 4% had elevated β2 microglobulin, and all patients were Binet A
*IgHV* the immunoglobulin heavy chain gene, *FISH* DNA fluorescence in situ hybridization, *ECOG* Eastern cooperative oncology group
[a]According to Dohner hierarchical Model
[b]Excluding del(17p)
[c]Excluding del(17p) and del(11q)
[d]no del(17p),del(11q),Trisomy12 and del(13q) for internal cohorts, and no del(17p),del(11q) and Trisomy12 for external cohort
[e]Excluding del(17p), del(11q), and trisomy12

top-ranked ensemble, CLL-TIM, was selected as the final model (Supplementary Fig. 4). We handled missing data using different methodologies (Methods subsection Handling of missing data). CLL-TIM is composed of 28 base-learners spanning both linear and non-linear algorithms. In total, CLL-TIM uses 85 original variables from patient histories (Fig. 2a), which translate to 228 engineered features (Fig. 2b and Supplementary Data 1). CLL-TIM also exhibited low redundancy among the selected features, where only 2% of all possible pair-wise feature correlations had an absolute Pearson's Correlation Coefficient (PCC) greater than 0.8 (Supplementary Fig. 5).

**Predictive uncertainty is reliably reported.** Central to achieving trust in an algorithm's predictions, is having an indication of when it might be making an erroneous prediction. A limitation in previous prognostic models in CLL[9–11,20–22] is the lack of uncertainty estimates for their predictions. The magnitude of agreement between CLL-TIM's 28 base-learners was used to indicate the confidence or uncertainty in CLL-TIM's predictions (Methods subsection Clinical trial requirements). This confidence was generated without any knowledge of the ground truth in patient outcome. On the separate internal test cohort (n = 646, Supplementary Fig. 1) that was used only after selecting CLL-TIM as our final model, CLL-TIM's predictions for patients meeting the high-confidence (HC) threshold were more accurate than for those with a low-confidence (LC) prediction (Supplementary Fig. 6, CLL-TIM HC, n = 261 Hazard-Ratio (HR): 7.3 (CI$_{95\%}$: 7.2–7.5) vs. CLL-TIM LC, n = 385, HR: 2.1 (CI$_{95\%}$: 1.8–2.8)). Thus, the magnitude of agreement between CLL-TIM's 28 base-learners was confirmed to be a reliable uncertainty estimate.

**CLL-TIM outperforms CLL-IPI and benchmark models.** To allow for benchmarking against the current gold standard CLL prognostic model, CLL-IPI[11] (developed for time to first treatment (TTFT) and OS), we defined an internal benchmark (BENCH-I; Supplementary Figs. 1, 4a) of 288 patients with both a full CLL-IPI and a full 2-year follow-up. On BENCH-I, CLL-TIM HC predictions (n = 145) achieved a precision of 0.72 (CI$_{95\%}$: 0.63–0.82) and a recall of 0.75 (CI$_{95\%}$: 0.65–0.86) resulting in a Matthews correlation coefficient (MCC) of 0.56 (CI$_{95\%}$: 0.42–0.70). This equated to a 2-year event-free survival (EFS) of 27.8 and 83.5% for the high- and low-risk group, respectively (CI$_{95\%}$: 18.4–41.9% and 76–91.8%, Fig. 3a, b and Supplementary Table 4). CLL-IPI score ranks a patient's risk with a score ranging from 0 to 10 (Supplementary Table 5). As MCC, precision and recall vary depending on the CLL-IPI threshold chosen, for an unbiased comparison to CLL-IPI, we used the threshold insensitive precision-recall area-under-curve[29,30] (PR-AUC) as a metric. CLL-TIM HC (n = 145) achieved a PR-AUC of 0.78 (CI$_{95\%}$: 0.69–0.86), significantly outperforming the highest PR-AUC achieved for CLL-IPI at 0.52 (CI$_{95\%}$: 0.43–0.61, p < 0.005 with one-tailed Mann–Whitney U test, Fig. 3c and Supplementary Tables 6–8). CLL-TIM also outperformed CLL-IPI for the 5-year composite outcome (p < 0.0005, Supplementary Tables 9–11). In addition, ensembles with variables selected by a data-driven strategy such as CLL-TIM, outperformed those using variables chosen by experienced CLL physicians and those not employing patient data prior to CLL-diagnosis (ENS-COMP$_{Doctor's Choice}$ and ENS-COMP$_{3-months}$, p < 0.05 with one-tailed Wilcoxon signed rank test, Supplementary Fig. 7).

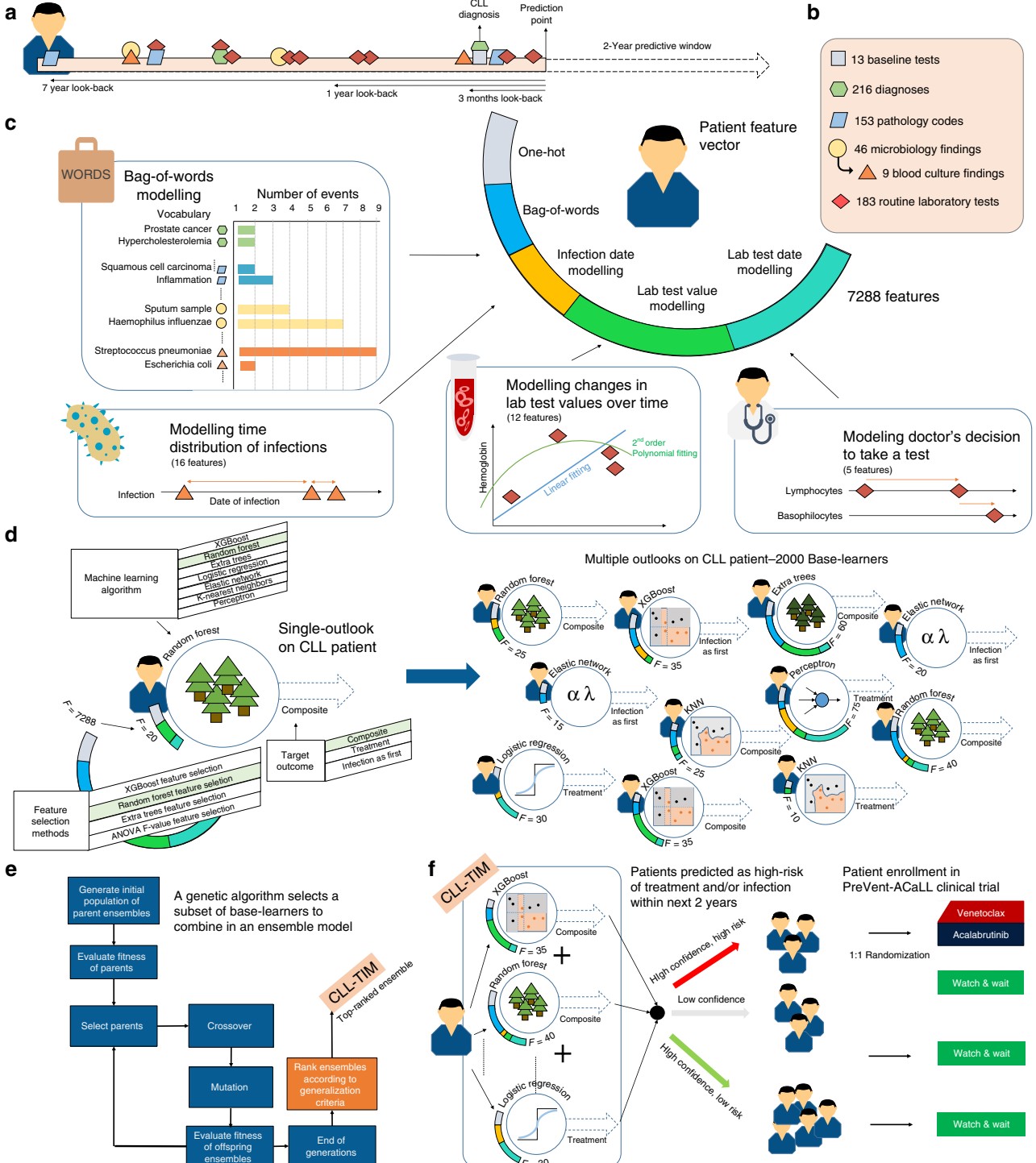

**Fig. 1 Development of CLL-TIM and selection of high-risk patients for PreVent-ACaLL clinical trial. a** For each patient, we modeled patient data in three look-back windows. Prediction-point was set at 3-months post-diagnosis and the 2-year risk of infection or CLL treatment (composite outcome) was the target outcome. **b** We assembled five datasets on 4149 CLL patients from the Nationwide Danish CLL registry, the Danish Microbiology Database, the Persimune data warehouse and health registries. **c** Using the Bag-Of-Words (BOW) approach, we modeled the frequency of occurrence of 216 diagnoses, 153 pathologies, and 46 microbiology findings (including 9 blood culture findings). We modeled the distribution of past infections and laboratory test results and designed latent features that model urgency of patient's condition and patient symptoms as interpreted by the treating physician. **d** Generation of a single base-learner (single outlook) required the random selection of: a machine learning algorithm; hyper-parameters; a target outcome and feature selection. In total, using this randomized protocol, we generated 2000 base-learners, each with their unique outlook into a patient's history. **e** A genetic algorithm (GA) was designed to generate 29 ensembles of 2–30 base-learners each. The generated ensembles were then post-ranked according to multiple criteria designed to maximize the generalizability of the ensemble. **f** The top-ranked ensemble chosen as CLL-TIM was then invoked to predict the 2-year composite outcome on a previously unseen test cohort and subsequently for the selection of high-risk patients for the PreVent A-CaLL Clinical Trial.

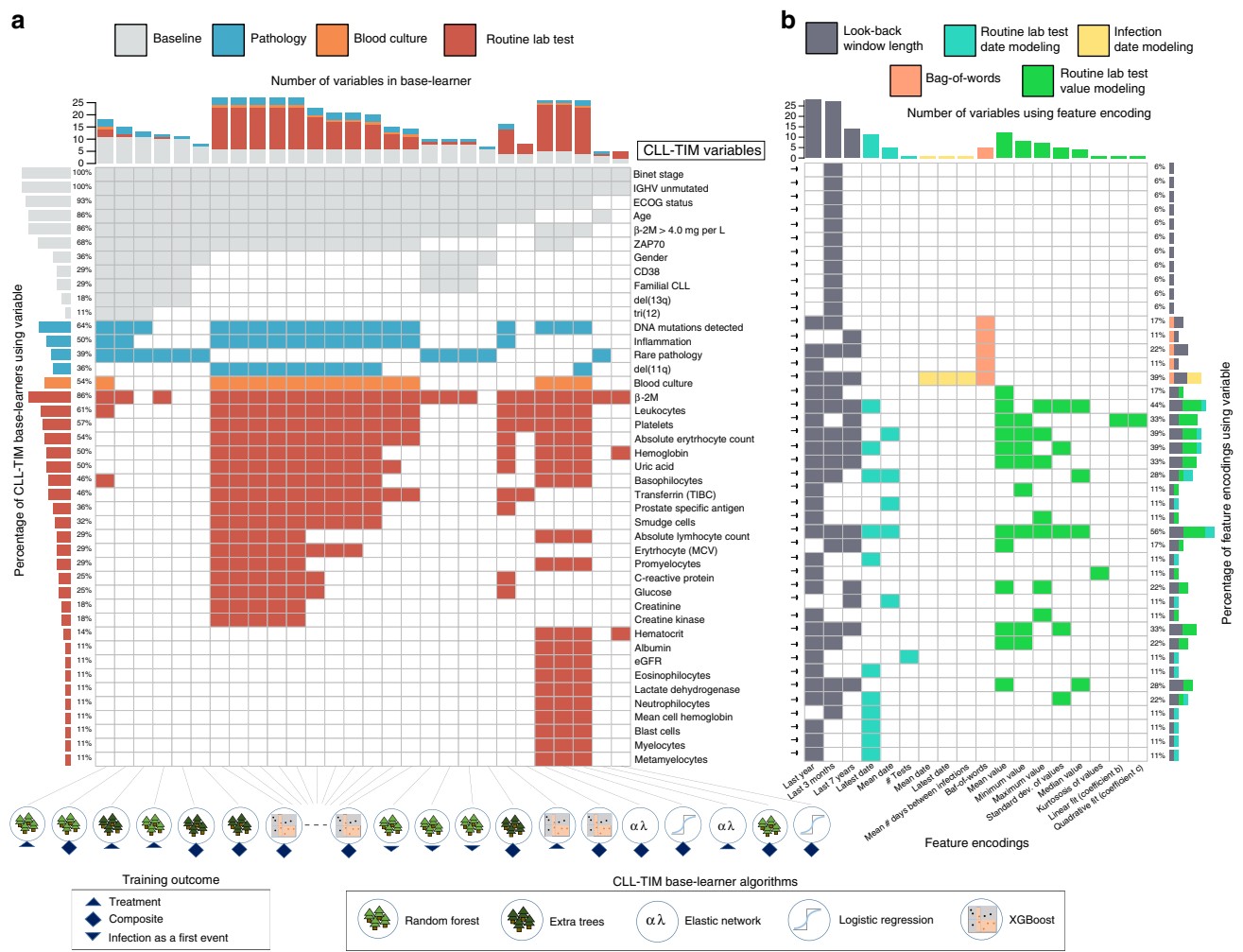

**Fig. 2 CLL-TIM ensemble composition: base-learners, variables and feature encodings.** CLL-TIM is composed of 28 base-learners: 20 model the composite outcome, 5 model CLL treatment outcome and 3 model infection as a first event. Base-learners include 13 XGBoost, 7 Random Forests, 4 Extra Trees, 2 Elastic Network and 2 Logistic Regression models. **a** Distinct variables used in CLL-TIM and **b** features that these variables are encoded into. Each variable may have multiple feature encodings (right panel in **b**) and can hence represent multiple features in the ensemble model. For instance, the variable Haemoglobin is encoded into four features; the mean, minimum and variability of the test result and the number of days since the last test. These four features were calculated on patient look-backs of 3 months, 1 year, and 7 years. In total, the feature encodings selected for 84 variables in CLL-TIM resulted in a set of 228 features (Supplementary Data 1). For visualization purposes, here we show a condensed representation with only variables that were used by at least 10% of CLL-TIM's base-learners (left panel in **a**). CLL-TIM models: rates of change, variability, average values and extremities in the results of several routine lab tests; recentness of lab tests and the number of tests taken (modeling doctor's decisions); several encodings representing the recentness and distribution of infection dates; counts of rare pathology codes (i.e. those found in <1% of the CLL training cohort) and inflammation events (pathology diagnoses). ECOG - Eastern cooperative oncology group, IGHV- The immunoglobulin heavy chain gene. B-2M - Beta-2 microglobulin, TIBC – Total iron-binding capacity, MCV – Mean cell volume, ECOG - Eastern Cooperative Oncology Group.

**Increased call rate for high-risk predictions with CLL-TIM.** The modeling of risk through the multiple outlooks of CLL-TIM's 28 base-learners (Fig. 2) was aimed at increasing the recall of high-risk predictions by allowing multiple paths in risk estimation of the composite outcome. On BENCH-I, CLL-TIM HC and CLL-TIM predicted 60 and 86 patients as high-risk, respectively ($n = 145$ and 288; Precision 0.72 (CI$^{95\%}$: 0.63–0.82) and 0.63 (CI$^{95\%}$: 0.55–0.72)). In turn, using CLL-IPI, 6 and 37 patients were respectively categorized as very-high-risk or above (7+), and high-risk or above (4+) ($n = 288$; Precision 1 (CI$^{95\%}$: 1–1) and 0.60 (CI$^{95\%}$: 0.45–0.74)). Additionally, 126 patients were categorized as intermediate risk or above (2+), but 54% of these patients had no infection or treatment within 2-years ($n = 288$; Precision 0.46 (CI$^{95\%}$: 0.40–0.52)). Furthermore, CLL-TIM enabled predictions on the remaining 48% of the test cohort patients, for which one or more CLL-IPI variables were missing (Supplementary Fig. 6e).

**Robustness to missing data on internal Danish test cohort.** CLL-TIM is composed of 85 variables encoded into 228 features (Supplementary Data 1), but designed to give robust estimations even when only a fraction of these variables are available. For instance, patients in the internal test cohort had a mean missing feature rate ranging from 3% to 48% per base-learner (Fig. 3d). To further test the robustness of CLL-TIM in potential real-world missingness scenarios, we re-tested CLL-TIM on BENCH-I by fixing entire variable categories to a missing value state. Robustness was claimed if CLL-TIM's performance in a missing data simulation, was still significantly

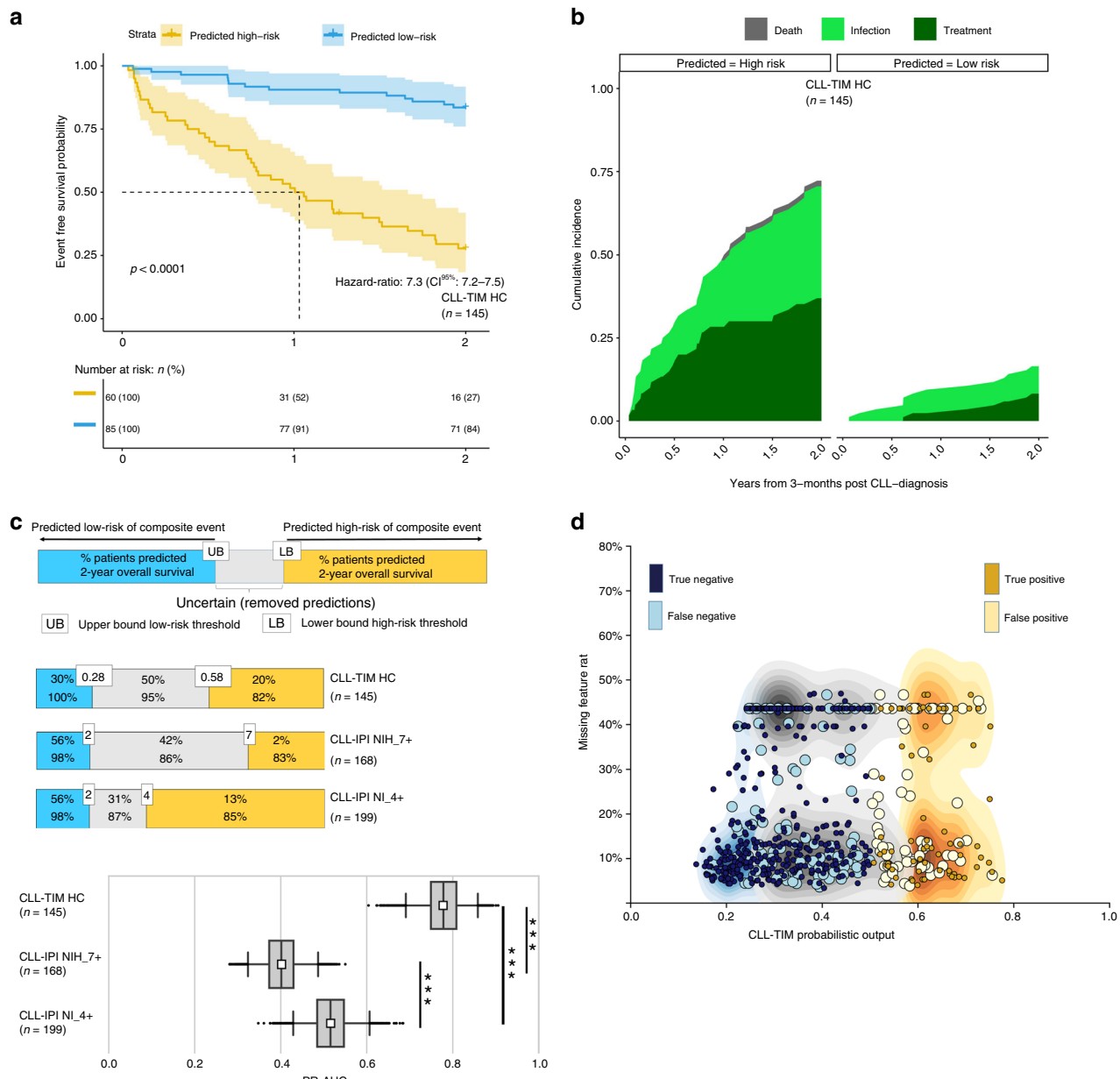

**Fig. 3 CLL-TIM benchmarking on internal test cohort. a** Kaplan–Meier graphs of infection-free, CLL treatment-free survival for CLL-TIM high-confidence (HC) predicted high-risk (yellow curve with 95% confidence intervals) and low-risk groups (blue curve with 95% confidence intervals) on a subset of the internal test cohort (BENCH-I, $n = 288$ with $n = 145$ high-confidence predictions). Patients in BENCH-I have full CLL-IPI and a full 2-year follow-up. $p$-value is by log-rank test. **b** Cumulative incidence plots for CLL-TIM HC predicted high-risk and low-risk groups for CLL treatment (dark green), infection (bright green) and death (grey) as first events on BENCH-I. **c** Two-year outcome PR-AUC (Precision-Recall Area-Under-Curve) for CLL-TIM and CLL-IPI on BENCH-I. To allow for an equitable comparison, CLL-TIM HC (i.e. with removal of uncertain predictions) was benchmarked against an additional two versions of CLL-IPI score; CLL-IPI with removal of patients in the intermediate-risk category, CLL-IPI NI_4+, and CLL-IPI with removal of the intermediate and high-risk category, CLL-IPI NIH_7+. For box-and-whisker plots, whiskers are 95% confidence intervals generated using 5000 bootstrapped datasets sampled from each respective cohort (See Methods), white square is the mean, centre line is the median, bounds of box are the interquartile range and black dots are outliers. We performed model comparison using a one-tailed Mann–Whitney U-test on the mean difference of the PR-AUC over the bootstrapped datasets. *** indicates $p < 0.0005$. **d** Average missing feature rate for patients in internal test cohort. Shaded distributions are blue – low-risk high-confidence predictions, gray – low-confidence predictions, gold – high-confidence high-risk predictions. Missing feature rate is the percentage of CLL-TIM's 228 features that were missing for the given patient. Data shown for CLL-TIM's predictions on the internal Danish test cohort ($n = 646$).

better than CLL-IPI. CLL-TIM met this robustness condition under the unavailability of all laboratory test variables, pathology data, blood culture data or diagnosis data; even when set to missing all together – hence, even on predictions for patients with only baseline data, CLL-TIM was still superior to CLL-IPI (Supplementary Fig. 8).

**Robustness to missing data on external CLL7 test cohort.** Next, we gathered data from the German phase 3 CLL7 watch and wait cohort (NCT00275054; $n = 365$ patients) as a real-world validation of CLL-TIM's robustness to missing data. This cohort included no data for infection, pathology, diagnosis and laboratory tests prior to CLL diagnosis and only 11 of CLL-TIM's 33

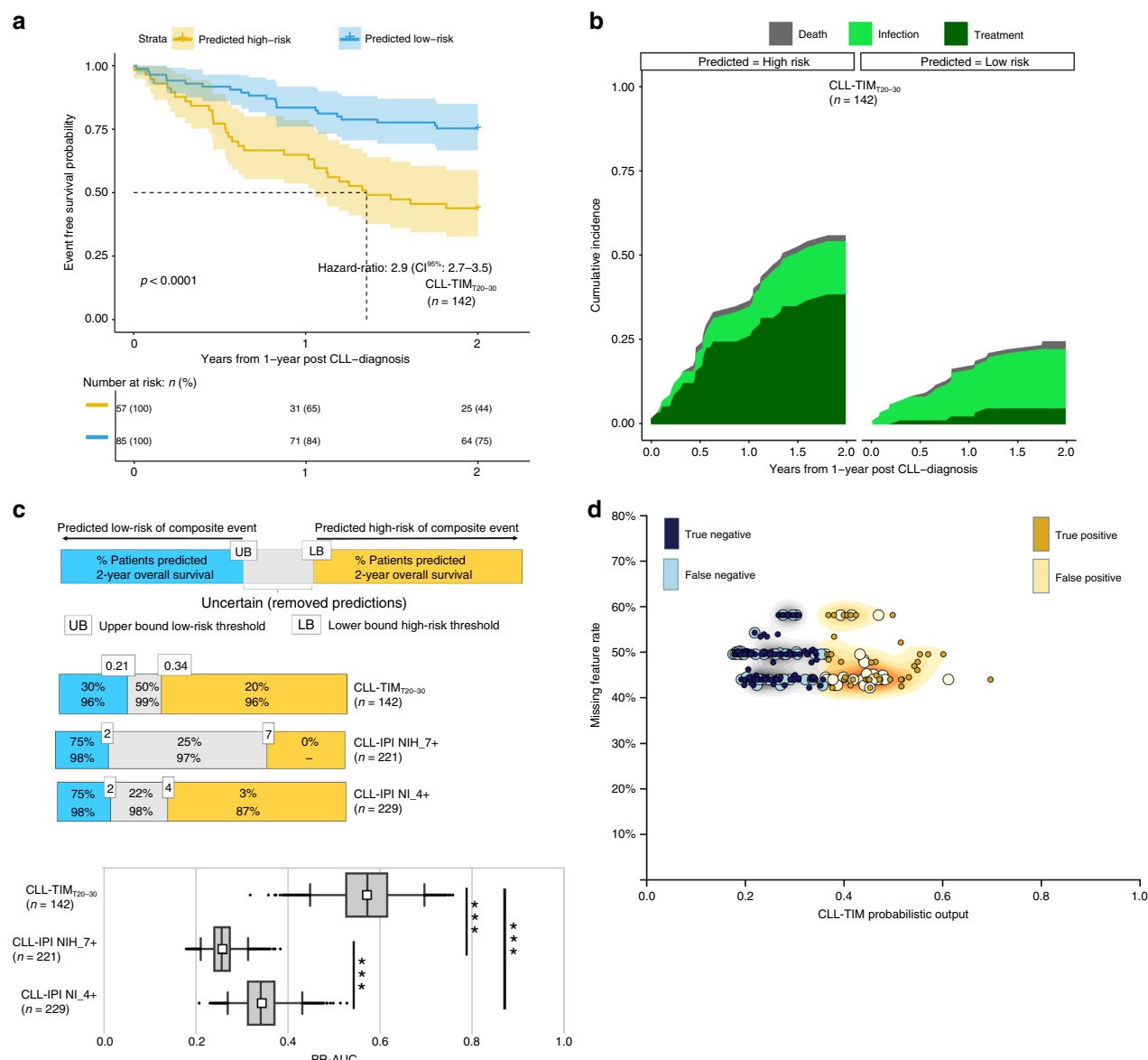

**Fig. 4 CLL-TIM is robust to missing data on external test cohort. a** Kaplan–Meier graphs of infection-free, CLL treatment-free survival for CLL-TIM$_{T20-30}$ predicted high-risk and low-risk groups on a subset of the external test cohort (BENCH-E, $n = 281$ with $n = 142$ high-confidence predictions). Patients in BENCH-E have full CLL-IPI and a full 2-year follow-up. *P*-value is by log-rank test. **b** Cumulative incidence plots for CLL-TIM$_{T20-30}$ predicted high-risk and low-risk groups for CLL treatment (dark green), infection (bright green) and death (gray) as first events on BENCH-E. **c** Two-year outcome PR-AUC (Precision-Recall Area-Under-Curve) for CLL-TIM and CLL-IPI on BENCH-E. To allow for an equitable comparison, CLL-TIM$_{T20-30}$ (i.e. with removal of uncertain predictions) was benchmarked against an additional two versions of CLL-IPI score; CLL-IPI with removal of patients in the intermediate-risk category, CLL-IPI NI_4+, and CLL-IPI with removal of the intermediate and high-risk category, CLL-IPI NIH_7+. For box-and-whisker plots, whiskers are 95% confidence intervals generated using 5000 bootstrapped datasets sampled from each respective cohort (See Methods), white square is the mean, centre line is the median, bounds of box are the interquartile range and black dots are outliers. We performed model comparison using a one-tailed Mann–Whitney U-test on the mean difference of the PR-AUC over the bootstrapped datasets. *** indicate $p < 0.0005$. **d** Average missing feature rate for patients in external test cohort. Shaded distributions are blue – low-risk high-confidence predictions, gray – low-confidence predictions, gold – high-confidence high-risk predictions. Missing feature rate is the percentage of CLL-TIM's 228 features that were missing for the given patient. Data shown for CLL-TIM's predictions on the external German test cohort ($n = 365$).

laboratory test variables (Supplementary Fig. 9). Baseline variables were all available except for del(13q). This resulted in an average missing feature rate ranging from 42 to 58% per base-learner; while the cohort overall had indolent baseline characteristics with half of the high-risk patients randomized for early treatment and thus removed prior to this analysis (Table 1). For comparison to CLL-IPI and BENCH-I, we considered 281 patients with a full CLL-IPI and a full 2-year follow-up (BENCH-E; Supplementary

Fig. 10c). Extracting the top 20% high-risk, and top 30% low-risk patients (T$_{20-30}$, Methods subsection Clinical trial requirements), CLL-TIM T$_{20-30}$ achieved a 2-year EFS of 43.9 and 75.3%, for those patients predicted as high-risk and low-risk, respectively (CI$^{95\%}$: 32.7–58.8% and 66.7–85.0%; Fig. 4a–c and Supplementary Table 12). Under these substantial missing data conditions, performance was still significantly better than CLL-IPI based on PR-AUC (Fig. 3c) and MCC (Supplementary Tables 13–18).

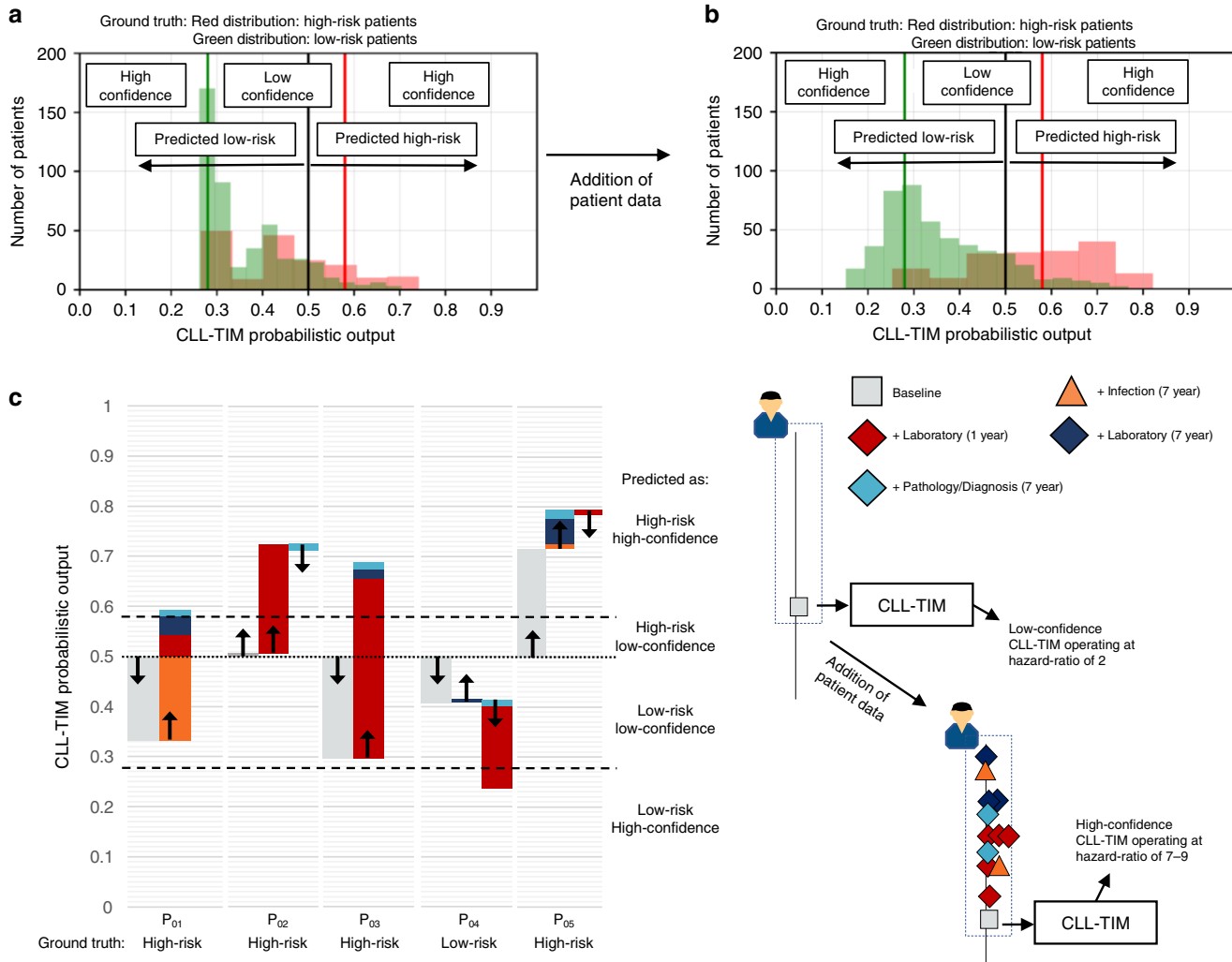

**Fig. 5 Addition of patient data increases CLL-TIM's performance. a** Distribution of CLL-TIM's probabilistic output when making predictions using only baseline variables. **b** Distribution of CLL-TIM's probabilistic output when adding patient data for 7-year histories of laboratory, infection, pathology and diagnoses. Predictions shown are for the internal test cohort, $n = 646$. Red distribution - truly high-risk patients, green distribution – truly low-risk patients. Overlaps of the two distributions represent misclassifications. Infection and laboratory histories had the most marked effects in increasing CLL-TIM's confidence in its predictions (See Supplementary Fig. 12). **c** Predictions on a subset of five patients ($P_{01}$–$P_{05}$) from the test cohort. $P_{01}$–$P_{03}$ and $P_{05}$ all had an infection or required treatment within the first two years from CLL diagnosis. $P_{01}$–$P_{03}$ were misclassified with CLL-IPI (CLL-IPI score of 1: low-risk). $P_{04}$ was a low-risk patient. We performed five rounds of CLL-TIM predictions on each patient, where in each round we added more data: baseline (gray); infection histories of seven years (orange); laboratory variables up to a one year prior to CLL diagnosis (dark red); laboratory variables up to a seven years prior to CLL diagnosis (dark purple); pathology and diagnosis variables (light blue). When predicting patients with only baseline variables, CLL-TIM only achieved low-confidence predictions for $P_{01}$–$P_{04}$. By adding more data on top of baseline variables, we were able correctly classify $P_{01}$–$P_{03}$ as high-risk with a high-confidence and $P_{04}$ as low-risk with a high-confidence. When CLL-TIM gives a low-confidence prediction, this means that with the given patient data, CLL-TIM is unsure of its prediction. In these situations, we can expect CLL-TIM to be operating at a Hazard-Ratio of around 2 (See Supplementary Fig. 6c, f). The benefit of CLL-TIM is that for low-confidence predictions, we may add more patient data to try and get predictions with a high-confidence. In this scenario CLL-TIM is more certain in its prediction and we can expect it to be operating at a HR of 7 or higher (Supplementary Fig. 6a, d).

## Addition of patient data reduces uncertainty in CLL-TIM.
As a result of the indolent baseline characteristics and missing data conditions of BENCH-E, we observed a reduction in performance in terms of recall (Supplementary Table 6 vs. 13) and the detection of infection events (Fig. 3b vs. 4b) when compared to BENCH-I. By re-simulating CLL-TIM's predictions on BENCH-I with similar indolent characteristics and missing data conditions as BENCH-E, we observed a similarly low recall, with a 2-year EFS of 61.5% and 72.1% for those patients predicted as high-risk and low-risk respectively (CI95%: 38.5–57.3% and 73.1–84.7%, Supplementary Fig. 11c). Upon including infection and pathology history prior to diagnosis and including data for more laboratory variables; we increased the detection of infections and recall by

3.4-fold and this with an improved discrimination between high- and low-risk groups — 2-year EFS of 40.4% and 78.7% respectively (CI95%: 43.8–86.5% and 66.4–78.3%, Supplementary Fig. 11e). The addition of patient data, also had the effect of increasing CLL-TIM's confidence in its predictions (Fig. 5a, b), thereby increasing the operating Hazard-Ratio of CLL-TIM (Supplementary Fig. 6). Predictions using only baseline variables, were of mostly low-confidence (Fig. 5a). Infection and laboratory histories had the most marked effects in increasing CLL-TIM's confidence, whilst pathology and diagnoses variables had more subtle effects (Supplementary Fig. 12). The addition of patient data from each variable category, affected the confidence for each patient differently (Fig. 5c).

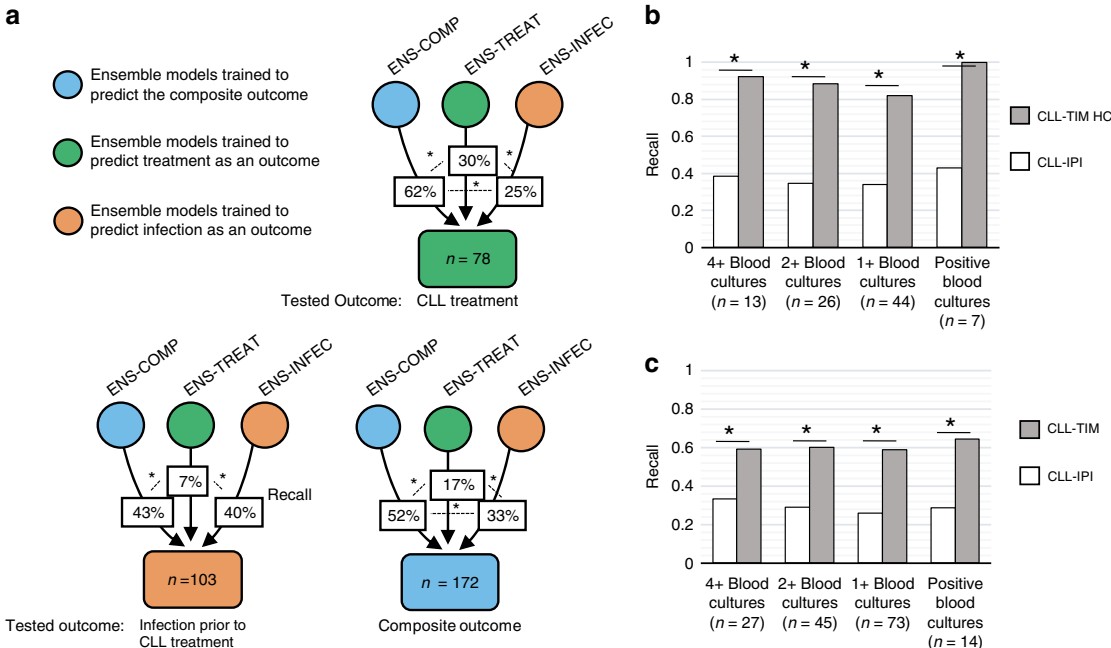

**Fig. 6 Linking immune dysfunction and CLL treatment. a** Modeling infection and CLL treatment separately and jointly. Results shown for patients with full 2-year follow-up on internal test cohort ($n = 530$). Percentages indicate the average recall resulting from 29 ensembles (2–30 base-learners) generated in each of the three protocols. The protocols differ in the target outcome they were trained to predict: ENS-INFEC (orange circle) predict infection as a first event; ENS-TREAT (green circle) predict treatment; ENS COMP (blue circle) predict the composite outcome. Irrespective of the outcome they were trained on, all protocols were tested for their ability to predict three different outcomes: Infection prior to CLL Treatment (with $n = 103$ patients having this outcome), CLL treatment ($n = 78$), composite outcome, which includes infection and/or CLL treatment ($n = 172$). The average recall for each set of ensembles represents the extent to which, patient data necessary for modeling one outcome (trained outcome) is predictive of another outcome (tested outcome). **b** Recall for alternative infection outcomes for CLL-TIM's high confidence predictions on BENCH-I ($N = 145$). In this work we use the drawing of a blood culture (1 + blood cultures), irrespective of the result, as a proxy for clinical infection. 4+ and 2+ are patients who had at least four and at least two blood cultures drawn within the first 2-years post CLL diagnosis, respectively. **c** Recall for alternative infection outcomes for all of CLL-TIM's predictions on BENCH-I ($N = 288$). For CLL-IPI score we used CLL-IPI 4+ where score of 4 and above is considered as a high-risk prediction and below 4 as low-risk. Only 10% of blood cultures taken in the Danish CLL population had a positive finding. Precision was not calculated for each of these four outcomes, as the false-positive predictions (according to each outcome) may still refer to patients who had CLL treatment. For CLL-TIM HC, CLL-TIM and CLL-IPI, precisions for the composite outcome were 0.72, 0.63, and 0.60, respectively. * indicates $p < 0.05$ for one-tailed binomial test for difference in proportions.

**Immune dysfunction linked to progressive disease**. To examine the link between infections and CLL treatment as outcomes, we compared ensemble models trained to predict the composite outcome to models that were trained to predict CLL treatment and models predicting infection as a first event (ENS-COMP, which includes the CLL-TIM ensemble; ENS-TREAT and ENS-INFEC, Supplementary Table 3). Training on infection as an outcome in combination with CLL treatment (as in ENS-COMP and CLL-TIM), synergistically improves the predictions of CLL treatment ($p < 0.05$ with one-tailed binomial test for ENS-COMP vs. ENS-TREAT, Fig. 6a). This result was corroborated by CLL-TIM also outperforming CLL-IPI in predicting CLL treatment on both internal and external cohorts (Supplementary Tables 19–24). Considering next, patients with infection prior to CLL treatment as an outcome, modeling only CLL treatment was not predictive (recall 7%, Fig. 6a). However, both ENS-INFEC and ENS-COMP models were predictive and had significantly similar recalls of around 40% ($p > 0.05$, Fig. 6a). In brief, modeling only CLL treatment was not enough to predict infection prior to CLL treatment, and modeling infection as an outcome, while necessary for the prediction of infections, was also somewhat predictive of CLL treatment (See Supplementary Fig. 13 for further evidence). Like ENS-TREAT, CLL-IPI was not predictive of infections prior to CLL treatment (Supplementary Fig. 14a, b). To rule-out whether this was due to CLL-IPI variables not being predictive of infection, or due to them not being trained to predict infection as

an outcome, we trained ensembles with CLL-IPI variables to predict both infection and treatment. Like CLL-IPI, these ensembles were also not predictive of infection (Supplementary Fig. 14c, d). We next assessed how the recall for infections differs when changing the definition for this outcome (Fig. 6b, c). While not statistically significant ($p > 0.05$), for high-confidence predictions, recall increased for patients with positive or multiple blood cultures. In turn, for all outcomes, CLL-TIM achieved significantly higher recalls than CLL-IPI.

**General and infection-specific risk factors**. To pinpoint features with impact on high-risk predictions, we calculated SHAP feature importance values[31] for all of CLL-TIM's 228 features and ranked them according to their ability to discriminate high- and low-risk patients (Supplementary Data 1, Methods subsection Risk factors). Features with most discrimination (referred to as risk factors) such as β2-microglobulin, Binet stage, and IGHV mutational status, corroborate previously described risk factors for CLL treatment and survival[11,32]. Several routine analyses also demonstrated good discrimination (Fig. 7a). These risk factors were separated into those specific to the detection of infection prior to CLL treatment and those specific to the detection of CLL treatment prior to infection (Fig. 7b, c). Here, we found that the mean number of days between infections (7-year look-back) was important for predicting infection prior to CLL treatment, as were

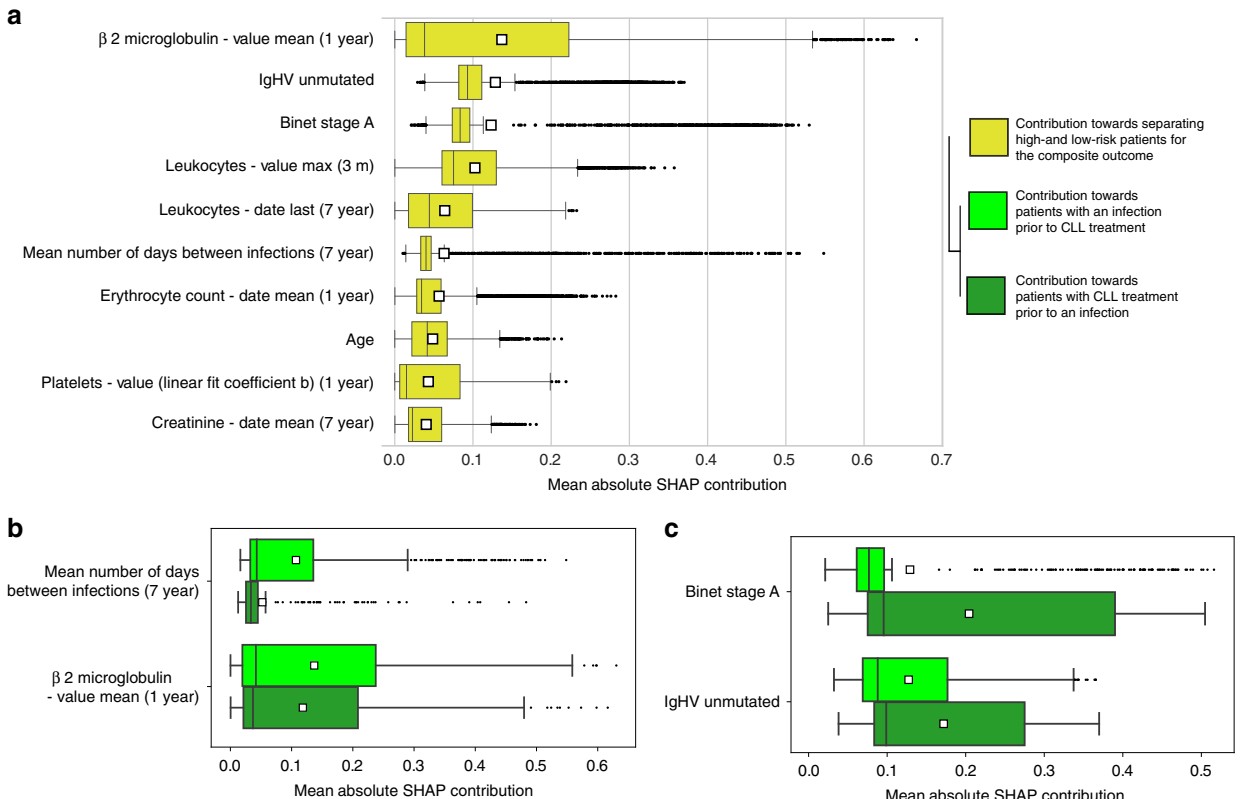

**Fig. 7 Risk factors for composite outcome, infection and treatment. a** Top 10 risk factors for the composite outcome. **b** Top 2 risk factors specific to infection prior to CLL treatment. **c** Top 2 risk factors specific to CLL treatment prior to infection. Mean absolute SHAP contribution indicates the magnitude by which the probabilistic output of CLL-TIM is affected by the given feature. This was calculated using 3720 Danish CLL patients (i.e. training, validation and test cohorts) and averaged over CLL-TIM's 28 base-learners. Specificity to infection in **b** was calculated as the mean difference between the feature's SHAP values for patients who as a first event, had an infection, to those that had CLL treatment. The converse of this was used to calculate specificity to CLL treatment in **c**. Mean differences for **b**, **c** were significant ($p < 0.005$) using one-tailed Mann–Whitney U-Test. Risk factors identified in a-c were also confirmed using multiple univariate tests (Supplementary Data 2, 3). For box-and-whisker plots, whiskers are the 95% confidence interval, white square is the mean, centre line is the median, bounds of box are the interquartile range and black dots are outliers. When patients had missing data for a given feature the SHAP contribution was zero.

several routine analyses (Supplementary Data 1). We observed that very few treatment-specific risk factors contributed to infection as a first event, while numerous infection-specific risk factors contributed to treatment as a first event (Supplementary Data 1). This is in line with our previous finding that while ENS-INFEC is predictive of treatment, ENS-TREAT is not predictive of infection. From univariate analysis of the 7288 features generated in this work, we found that for CLL-TIM's top 50 features, 92% were significant ($p < 0.05$) in at least one univariate test, while 72% were significant ($p < 0.05$) in all four univariate tests (Supplementary Data 2). We also observed homogeneity for the features exhibiting the lowest p-values with each univariate test (Supplementary Data 3). These consisted of features based on clinical stage, β2-microglobulin, hemoglobin and leukocytes. We also performed the same univariate analysis but with infection (as a first event) as the only outcome (Supplementary Data 4). Here, all four univariate tests were dominated by infection related features. Composite outcome risk factors (See Fig. 6a) were confirmed significant in all four univariate tests, except for three that were significant in only two of the four univariate tests. From the 15 infection risk factors (Supplementary Data 1), 13 were significant in two or more univariate tests (Supplementary Data 4). Visualization of key risk factors identified by CLL-TIM, showed that CLL-TIM was able to learn complex non-linear functions (Fig. 8). For instance, depending on the value of other features, similar levels of β2-microglobulin may contribute very

differently towards risk of a composite event. This idea of conditional risk contrasts greatly to the independent and additive contributions of factors in prognostic indices such as CLL-IPI, with β2-microglobulin either contributing two or zero points[11]

**Personalized risk factors and multiple etiologies of risk.** While risk factors on a population level encapsulate tendencies in a CLL population, a next step towards personalized medicine is the derivation of risk factors specific to each patient that drive the prediction in a given state. As part of a web-based beta version of CLL-TIM (Fig. 9a, CLL-TIM.org), available for prediction of individual patient outcomes, we provide personalized risk factors driving the patient into high- or low-risk and a confidence value for the specific prediction (Fig. 9b). To analyze the variability, if any, in the personalized risk factors for patients with a composite outcome, we extracted, from the personalized risk factors of each, the top 3 factors pushing the patients towards high-risk (Fig. 9c). Found in 14% of the patients, the most common combination of risk factors was Binet stage, IGHV mutation status and leukocytes maxima in the last 3 months. For 50% of the population, 17 different combinations of 12 risk factors were found in the top 3 personalized risk factors. For characterizing the risk of the remaining 50% of the patients, 313 combinations of 34 risk factors were in turn required – thus highlighting the strength and necessity of having a multiple-outlooks strategy with a

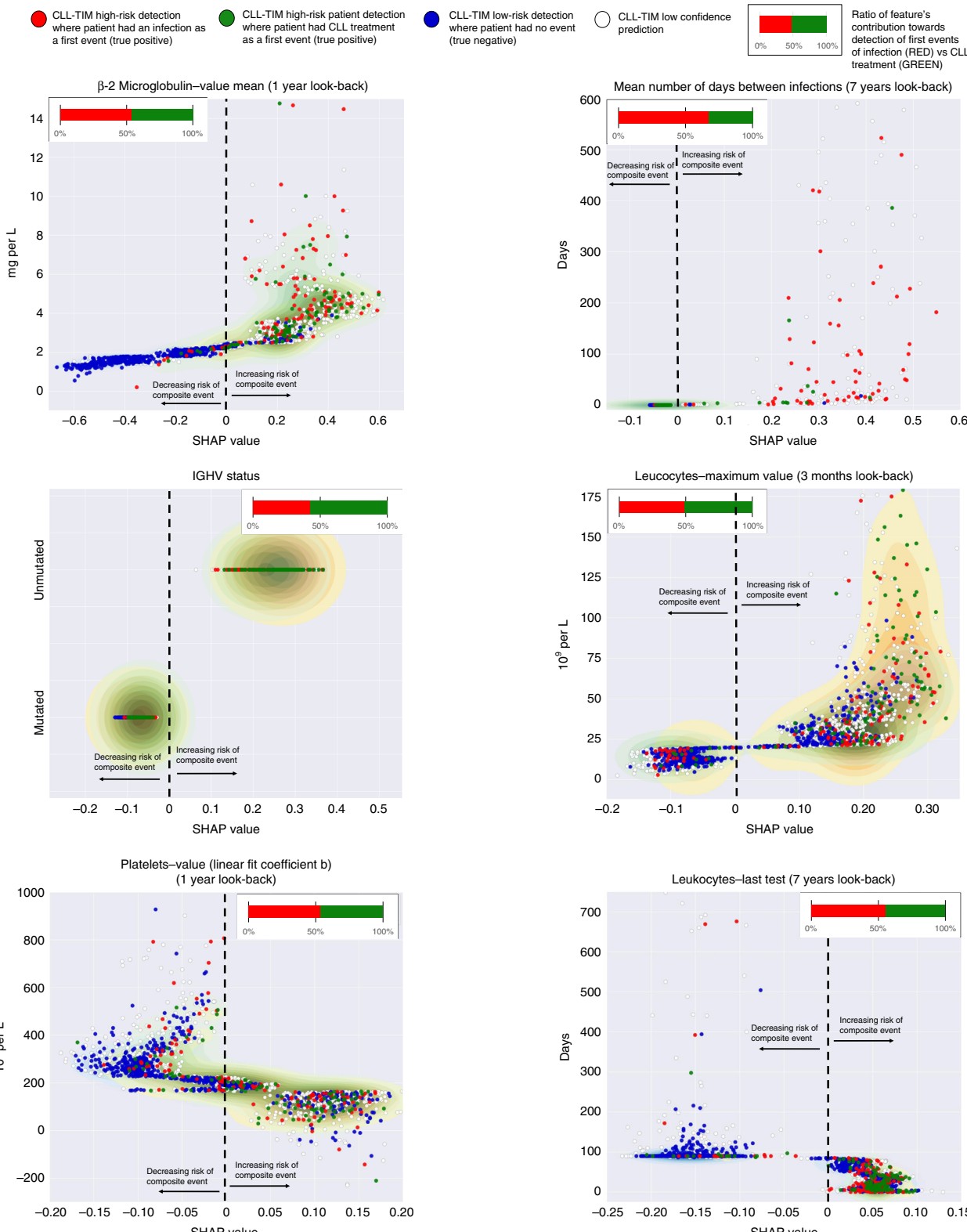

**Fig. 8 Non-linear relationships between risk factors and the composite outcome.** Variable-to-risk mappings learned by CLL-TIM are presented for a selection of key risk-factors. Red and green circles are for high-confidence true-positive predictions were patients had an infection and CLL Treatment as a first event respectively. Dark blue circles are for high-confidence true-negative predictions. Low-confidence predictions are shown in white circles. On the y-axis the real feature value for the given patient is shown and, on the x-axis, the respective SHAP feature value for the given patient is shown. Plots were generated using 3720 Danish CLL patients (i.e. training, validation and test cohorts).

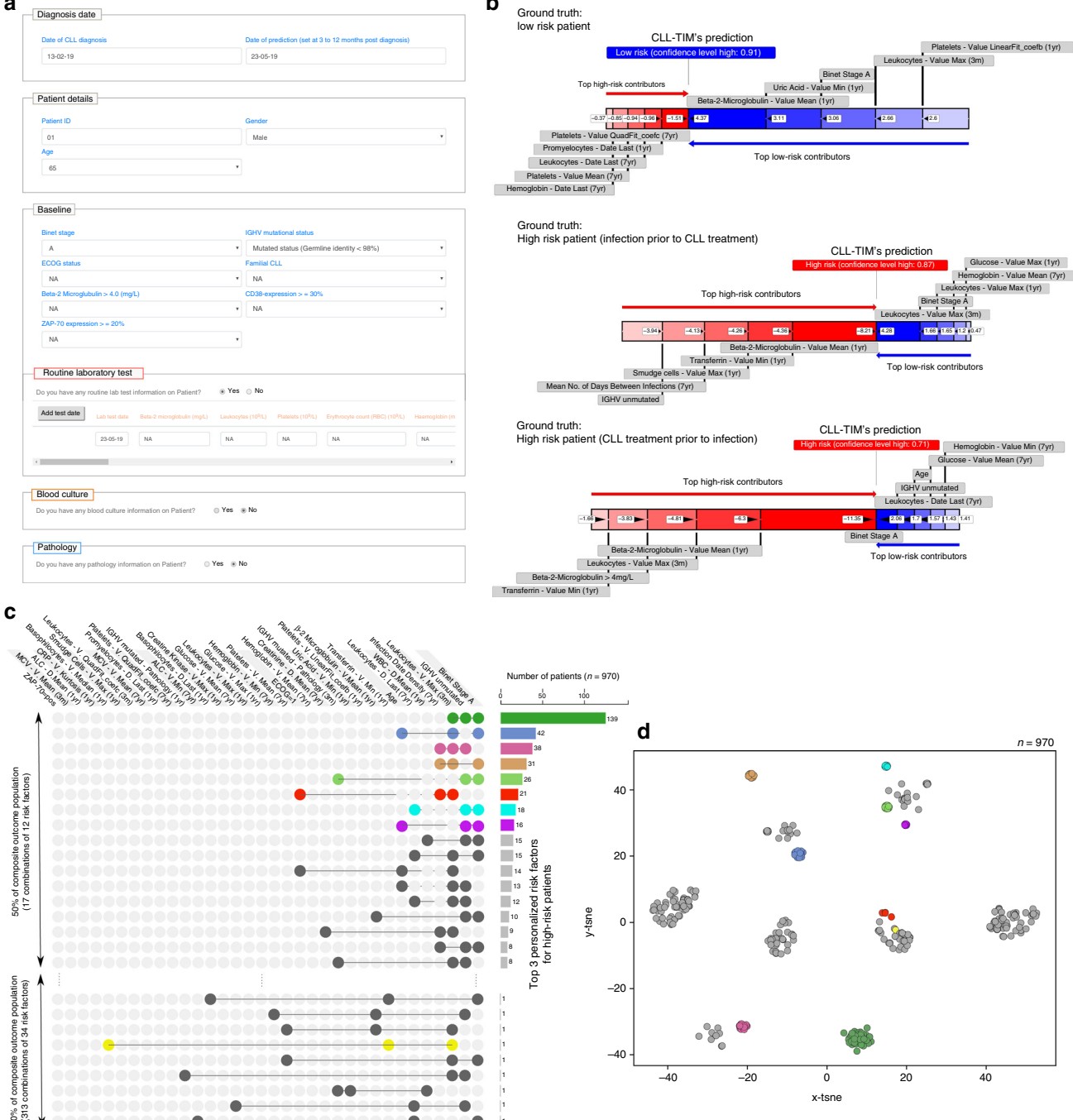

**Fig. 9 Analysis of personalized risk factors shows multiple paths towards risk of a composite event. a** Input interface of CLL-TIM webserver. CLL-TIM may be used with any patient data available and after data input, the physician is provided with: the predicted risk (high or low) of a composite event within the next 2 years; a confidence value indicating the reliability of the given patient prediction; a set of risk factors specific to that patient that were responsible in driving the patient towards their predicted risk. **b** Personalized risk factors provided by CLL-TIM's webserver. Examples are for three patients in the internal test cohort for which CLL-TIM correctly predicted the outcome of, with high confidence. CLL-TIM's probabilistic outputs were 0.09 (low-risk, high-confidence) for the first patient, and 0.87 and 0.71 (high-risk, high-confidence) for the remaining two patients. To calculate these risk factors, for each patient, the average SHAP value for each of CLL-TIM's 228 features was extracted and averaged across the 28 base-learners of CLL-TIM. The features were then positively and negatively ranked to extract the top 5 features pushing the patient towards high-risk and those pushing the patient towards low-risk. **c** Co-occurrence of the top personalized high-risk factors. All features that were in the top 3 personalized high-risk factors for the 970 patients in the entire Danish cohort with a composite outcome. In total, 970 high-risk patients had 330 unique combinations of 34 features in their top 3 personalized high-risk factors. Figure generated using UpSet web-interface. **d** t-distributed Stochastic Neighbor Embedding (t-SNE) clustering of the top 3 personalized high-risk factors. t-SNE was performed using scikit-learn with default parameters on the top 3 personalized high-risk factors for 970 patients with a composite outcome. Color coding is according to **c**.

heterogeneous set of features. For the 970 patients in the Danish cohort with a composite outcome, we performed clustering of the top 3 personalized risk factors with Stochastic Neighbor Embedding[33] (t-SNE – Fig. 9d) and Uniform Manifold Approximation and Projection[34] (UMAP - Supplementary Fig. 15). Results from t-SNE and UMAP show that patients have distinct clusters of personalized risk factors. Similar to previous comparisons[35], UMAP clusters were more defined and less diffused than those with t-SNE.

## Discussion

While infection is the major cause of death in CLL[1], no prognostic index for prediction of infection in CLL has been presented prior to our recent work[36]. A first step in attempting to change the ramifications of immune dysfunction in CLL is to identify high-risk patients prior to any infection or CLL treatment[5]. We addressed this unmet need by developing an explainable machine learning model based on data from 4,149 patients diagnosed with CLL in Denmark between 2004 and 2017. CLL-TIM identifies patients at risk of severe infection or CLL treatment within 2-years of CLL diagnosis with high precision (0.72 CI$^{95\%}$: 0.63–0.81) and recall (0.75 CI$^{95\%}$: 0.65–0.86). Higher PR-AUC, MCC and a clearer discrimination for 2-year EFS between high- and low-risk patients were achieved as compared to all previous models. CLL-TIM predicted infection and CLL treatment with similar frequencies as first events, thus validating the ability of CLL-TIM to identify both patients at risk of infection and in need of CLL treatment. Predictions on the entire test cohort ($n = 646$) are representative of the Danish CLL population as CLL-TIM is modeled upon a registry with 98% coverage of patients diagnosed with CLL in Denmark[37]. However the Danish cohort represents a more indolent population than in most other published cohorts[11,37]. Patients with incomplete diagnostic work up fare worse than patients with full diagnostic work up[38]. This may affect predictions on the subset of patients with full CLL-IPI (BENCH-I, $n = 288$ and BENCH-E, $n = 281$ cohorts). Bootstrapping was necessary for MCC and PR-AUC, as CIs for these metrics cannot be automatically generated by using the gaussian approximation[39]. One limitation of comparing model performance on a single test set using the bootstrap method[40,41], is that for small sample sizes, CIs were sometimes wide and overlapping. Particularly, when low-confidence predictions were included. However, even in these conditions, CLL-TIM still achieved significant performance improvements over CLL-IPI.

In this work we find that using the drawing of a blood culture as a proxy for serious clinical infection holds both prognostic (Fig. 6b, c and Supplementary Data 1 and 4) and learnable information (Fig. 6a and Supplementary Table 6). This agrees with guidelines and standard practice to draw a blood culture upon suspicion of a serious infection. We cannot rule out however, that for some infectious events, we may be annotating events that have similar symptoms to infection but do not represent true infections, as is also the case in the clinical setting. The increased risk of mortality for CLL patients with infections during the first year from diagnosis, has been demonstrated both when using a blood culture as a proxy for infection[36] and when using infections classified as serious by physicians[7,42]. Although we achieved a 100% recall for positive blood cultures in CLL-TIM's high-confidence predictions, the patient count was too low for any significant differences to be observed between prediction of blood cultures with any finding and positive blood cultures. A link between immune dysfunction and CLL aggressiveness has also been established in our data driven approach. Analyses aimed at understanding the link between the risk of infection and risk of treatment showed that modeling infection together with CLL

treatment was necessary for the detection of infection prior to CLL treatment but also improved the detection of CLL treatment prior to infection (Fig. 6a). Thus, by combining the two clinically interlinked outcomes of immune dysfunction leading to risk of infection and risk of CLL treatment into a joined outcome, we were able to draw mutually predictive information from both event types. This also improved predictions for both outcomes. The correlation between immune dysfunction in CLL and aggressive disease is in accordance with recent reports of inferior survival and TTFT for CLL patients suffering from infection within the first year of diagnosis[6]. The link between immune dysfunction and aggressive or progressive CLL is further supported by the microenvironmental interaction being fundamental for CLL development[13–16]. CLL-IPI was developed for prediction of OS and validated for TTFT but performed poorly for prediction of infection. This is in line with our own ensemble models that were trained only to predict CLL treatment not being predictive of infection. Additionally, for ensemble models using only CLL-IPI variables but re-trained to predict the composite outcome, prediction of infection prior to CLL treatment was obfuscated with false-positives, resulting in low precisions. Thus, approaching CLL modeling with a larger and more heterogeneous set of features, together with the addition of infection as an outcome, were necessary to improve overall detection.

Recent recommendations for machine learning implementations in clinical practice propose interpretability of predictions along with robustness to incomplete data as the hallmarks of a clinically viable model[43]. Whereas 'opening the black-box' by explainable predictions as provided by CLL-TIM is key to gaining trust in the clinical setting, we believe that an additional — somehow overlooked — requirement, is the justification for using a complex model. Based on a multiple-outlooks ensemble model of 28 machine learning algorithms, trained on different feature sets, CLL-TIM is inherently complex. From one standpoint, the complexity can be justified as being more conducive of the reality of CLL — a complex disease ranging from an indolent to a rapidly fatal disease course in which tumor microenvironmental interaction, immune dysfunction, prior infections, and comorbidities all impact the clinical course[1,7,17,36,44,45]. From an equally pertinent stand-point however, we propose that the multiple-outlooks strategy used for the development of CLL-TIM increases base-learner diversity[46–48], and hence by design, is the main driving force behind CLL-TIM's (i) robustness under high-rates of missing data[40,49], (ii) ability to generate reliable uncertainty estimates for its predictions[50], (iii) improved recall[51], and (iv) ability to generate personalized risk factors. Considering the real-world data in the Danish CLL registry, CLL-IPI could only assess risk for 48% of patients due to missingness. In contrast, CLL-TIM outperforms CLL-IPI and can assess risk for any CLL patient even without information for most variables. With a significant increase in MCC and PR-AUC, CLL-TIM improved the recall as compared to CLL-IPI without jeopardizing precision.

In line with work deriving uncertainty estimates from ensembles[52,53], CLL-TIM's uncertainty estimates allows the treating physician to assess whether a predicted risk is actually of clinical significance for the patient. As exhibited by the low HR for low-confidence patients (Supplementary Fig. 6c, f), a low-confidence prediction means that for this patient, CLL-TIM cannot give a more accurate prediction. For a high-confidence prediction, CLL-TIM is more certain of its prediction and operates at a HR of 7–9 (Supplementary Fig. 6a, d). A related benefit of CLL-TIM is that upon input of further patient data, low-confidence predictions may still be converted into high-confidence predictions (Fig. 5). For the external German CLL7 validation cohort, we observed a lower call-rate of high-risk predictions both by CLL-IPI and CLL-TIM. Partly explaining this, is

that half of the high-risk patients in the external German CLL7 validation cohort were randomized for early treatment and not part of our predictions. Additionally, the external cohort has limited data besides baseline variables, which we have shown to have very limited separability of high-risk and low-risk patients (Fig. 5a). Further corroboration to this was found through simulations on the internal cohort, where certain high-risk patients could only be correctly identified with the addition of infectious history and laboratory variables. Similarly, upon performing simulations on our internal cohort that mimic the external cohort, both recall of high-risk predictions and detection of infections was reduced when using only post-diagnosis data. This, especially for those patients with indolent baseline characteristics. Given that current prognostic models in CLL are based on a handful of variables around time of diagnosis or treatment[9–11,20–22,54], for future modeling strategies, we recommend inclusion of laboratory values (without categorical cut offs) and medical history including infections pre-dating diagnosis — most particularly for those models addressing immune dysfunction.

In contrast to models that simply add the impact of individual risk factors[9–11,20–22], we uncovered the more complex and non-linear mechanisms by which prognostic factors contribute toward composite risk in CLL (Fig. 8 and Supplementary Fig. 16) — this, both on a population (Fig. 7a and Supplementary Data 4) and individual level (Fig. 9b, c). Our data-driven approach also identified novel prognostic factors for immune dysfunction in CLL (Fig. 7b), and did not select for immunoglobulins (Supplementary Discussion). We demonstrated that the density of infections several years prior to CLL diagnosis was highly specific for the risk of infection after diagnosis but prior to CLL treatment. As infections remain the major cause of death in CLL[6], with increased incidence after chemoimmunotherapy[2,3,8], identification of patients at risk of infection and the provision of personalized risk factors can aid efforts aimed at personalized treatment for CLL. The ability to derive personalized risk factors in CLL-TIM is a culmination of efforts in data cleaning, validation, and integration by PERSIMUNE[55], combined with a diverse modeling approach developed by the multiple perspectives from a team of physicians, molecular biologists, bioinformaticians, and data analysts during MispCamp[23]; and, the use of state-of-the-art developments in interpretable machine learning tools[31,56]. We also expect that PERSIMUNE data lake will provide the basis for automating and integrating CLL-TIM directly into the medical record, further integrating medical AI.

The next step is using CLL-TIM for patient selection in the investigator-initiated, randomized phase 2–3 trial within the Nordic and the Hovon CLL study groups, the PreVent-ACaLL trial (clinicaltrials.gov: NCT03868722). This trial investigates whether three months of acalabrutinib and venetoclax combination targeted treatment can improve the grade ≥3 infection-free, treatment-free survival compared to the standard-of-care observation arm of the study. Addressing the lack of prospective validation for machine learning in the clinic[57], through the PreVent-ACaLL trial we will also be able to further assess the predictive performance of CLL-TIM, as the observation arm and the patients predicted with low risk or low confidence predictions will be followed for infections and CLL treatment. To allow assessment of CLL-TIM in clinical practice in the meantime, we have made the model publicly available as a web-based beta version that includes confidence estimates and personalized risk factors (CLL-TIM.org).

## Methods

**Data sources**. To model multiple aspects of each patient, we assembled data from several sources (Fig. 1a) into five datasets. (1) Baseline variables taken at time of diagnosis (±3 months). We extracted these variables from the Danish National CLL registry and include age, gender, Binet stage, family history of CLL, Eastern Cooperative Oncology Group (ECOG) performance status, β2-microglobulin levels, CD38 positivity, 70 kDa ζ-associated protein (ZAP70) positivity, IGHV status, FISH for del(13q), tri(12), del(11q) and del(17p) (Supplementary Table 25). (2) Routine lab; Routine laboratory tests including all available biochemical blood work-up such as hemoglobin levels, complete blood counts, and C-reactive protein levels. (3) Microbiology including blood culture findings from the Persimune Microbiology Database[55]. (4) Pathology reports (SNOMED codes used as variables) from the Danish Patobank. (5) Diagnosis codes from the Persimune Data Warehouse[55] were used as variables, i.e. International Classification of Diseases (ICD-10 codes). In order to reduce potential bias when comparing our model to the CLL-IPI, we did not include CLL-IPI[11] scores as a variable. To keep the process entirely data-driven for routine laboratory tests, microbiology, pathology and diagnosis codes, we used all available patient data for modeling. Hence, any variables from these data sources that were included in the final CLL-TIM model, are, because of data-driven methods, described in the following sections. The study was approved by the Danish National Committee on Heath Research Ethics and informed consent was not required for retrospectively included patients according to Danish legislation.

**Base-learner generation**. In this work, we employ an ensemble modeling approach, which is the combination of predictions from multiple classifiers to produce a single classifier[58]. The aim was to make the ensemble predictions more accurate than any of the individual classifiers included in the ensemble. This requires accurate ensemble classifiers (referred to as base-learners in context of ensembling) and uncorrelated predictions, i.e. allowing for errors on different examples[59]. For instance, it has been shown that combining base-learners with independent errors and with accuracy that is only slightly better than chance, as the number of classifiers combined in an ensemble goes to infinity, the error can be reduced to zero[59]. In addition, the attractive property of reducing overfitting through ensembling[46] is integrated in the design and success of recent popular machine learning methods like XGBoost and Random Forest[60,61]. Although complete independence between the base-learners is hard to achieve, there are many ways to reduce correlation between base-learner predictions[47]. The entire protocol for CLL-TIM: feature generation (described in Supplementary Methods subsection Feature Generation); base-learner generation, ensemble generation and ranking (described in the forthcoming Methods sections), were motivated by producing a final ensemble with high accuracy and low correlation between its base-learners. Aimed at generating a diverse set of base-learners, we trained 2,000 base-learners on the training cohort, where the base-learners spanned seven types of linear and non-linear machine learning classifiers and were trained using different hyper-parameter settings, feature sets and target outcomes (Fig. 2d and Supplementary Table 3). The protocol used for base-learner generation was as follows: (i) Assignment of Target Outcome: 1000 base-learners were assigned to predict the composite outcome of risk of infection or CLL treatment: 500 base-learners were assigned to the CLL treatment outcome and 500 base-learners to the infection as a first event outcome. (ii) Assignment of Algorithm Type: An algorithm was randomly assigned from a choice of seven different classification algorithms (K-Nearest Neighbors (KNN), Logistic Regression (LR), Elastic Network (EN), Perceptron, Random Forest (RF), extremely randomized Trees — Extra Trees (ET), and Extreme Gradient Boosting — XGBoost (XGB)). The choice of these specific seven algorithms was based on their ability to model the full range of linear to non-linear decision boundaries, their ability to produce a confidence value associated with their prediction, and efficient training. All algorithms were implemented using Scikit-Learn[62] and the XGBoost library[61]. (iv) Assignment of Hyper-Parameter Settings: Hyper-parameter settings were assigned randomly from the ranges detailed in Supplementary Table 3. To maximize de-correlation between base-learners of the same algorithm type, the hyper-parameters were not optimized and remained as per their initial random assignment. (v) Feature Bagging: To reduce the feature set size from 7288, we employed a number of feature selection methods to rank features according to their importance in separating high-risk and low-risk patients (Supplementary Table 3). We then trained the base-learners using a subset of the top ranked features, where the feature set size was a randomly assigned integer ranging from 7 to 150. This is effectively a method of feature bagging[63] that promotes feature diversity in base-learners but is still biased toward the top ranked features. (vi) After the assignment of the target outcome, algorithm type, hyper-parameters and feature set, we trained the base-learner on the training cohort. We performed five repetitions of 10-fold cross-validation — identical folds were used for different base-learners. We performed feature selection separately within each fold to avoid data leakage across folds and overestimation of cross-validation performance. For quality control, we then pre-filtered the set of 2000 base-learners to remove low performing base-learners. Base-learners with cross-validation Matthews Correlation Coefficient (MCC) smaller than the respective base-learner population mean, were not put forward for selection.

**Ensemble generation**. After generating a set of diverse classifiers, each with their unique outlook into the patients' history (Fig. 2d), we subsequently devised a method to solve the combinatorial optimization problem of finding a subset of base-learners that could maximize the MCC on the validation cohort (Supplementary Fig. 1). To this end, we used a meta-heuristic optimization algorithm

inspired by natural evolution called Genetic Algorithm (GA)[64]. The details for a single GA run are provided in Supplementary Fig. 17 and summarized in Fig. 2e. The GA was run 29 separate times to generate ensembles ranging from 2 to 30 base-learners.

**Ensemble ranking.** To select a single ensemble that was most likely to generalize well on unseen test cohorts, the 29 ensembles generated by the GA required a model selection procedure. Selecting as the final ensemble, the ensemble that achieves the highest MCC on the validation cohort does not necessarily guarantee good generalization to an unseen test cohort. This because given that a large number of ensembles are assessed by the GA for their performance on the validation cohort, there is the risk of over-fitting to the validation cohort. Therefore, apart from the ensembles' performance on the validation cohort, we used six criteria for ensemble ranking that were designed to maximize the generalization ability of the ensemble (Supplementary Table 26). These criteria promote accuracy, predictive and compositional dissimilarity between the base-learners. The score of each ensemble was calculated as the average score over the six criteria after standardization of each. We then ranked all 29 ensembles (of sizes 2–30 base-learners) according to this score and the apex ensemble (CLL-TIM) was then tested for its generalization performance on internal and external cohorts (Table 1).

**Handling of missing data.** Missing data was handled over several layers in the design of CLL-TIM. On the feature level, baseline variables were one-hot-encoded, hence we have a feature for each baseline variable indicating when it is missing. For the laboratory data, we created features (Lab Test Date Modeling Variables; Supplementary Table 1) that are missingness indicators. In this way the learning algorithms, through these features, can take into account recentness of the test dates or the lack there-of. On the base-learner level, XGBoost[61] is capable of creating splits for missing data without imputation. For the other methods, we used median imputation for their features. No imputation was performed for baseline variables as these are one-hot-encoded. On the ensemble level, the ensemble was designed in such a way to promote diversity in predictions (Methods subsections Base-learner generation and Ensemble ranking), the rational being that different parts of the ensemble may be able to compensate for missing data in others[40,49]. Given that we take the average probabilistic output over all base-learners, this means that we are implicitly down-weighting base-learner predictions with missing data.

**Benchmark models.** CLL-TIM was benchmarked against the current gold standard in CLL prognostication, CLL-IPI[11,65]. The protocol used to generate CLL-TIM (i.e. data-driven strategy that models patient data prior to CLL diagnosis and predicts the composite outcome) was compared to five other protocols of generating ensembles. Details and motivations of all benchmark models and protocols are provided in Supplementary Table 5. The protocols include: ensembles trained to predict CLL treatment as an outcome and those that predict infection as an outcome. This enabled us to evaluate the effect of modeling the composite outcome of CLL treatment and infection, both separately and jointly. In addition, other protocols included: ensembles comparing the data-driven approach of CLL-TIM to that of two experienced physicians pre-selecting variables; ensembles comparing the effect of modeling data prior to CLL diagnosis; and ensembles using only CLL-IPI variables that are trained to predict the composite outcome.

**Evaluation metrics.** We assessed the 2-year composite outcome of risk and/or infection using Kaplan–Meier and Aalen-Johansen plots. As 26% of the patients in the training cohort had the composite outcome (Supplementary Fig. 1), the class-imbalance renders common evaluation criteria such as accuracy and the Area-Under-Curve (AUC) of the receiver-operating-characteristics (ROC) to have overoptimistic results[30]. We thus opted for metrics that are insensitive to class-imbalance, namely the Matthews Correlation Coefficient (MCC):

$$MCC = \frac{TP \times TN - FP \times FN}{\sqrt{(TP + FP)(TP + FN)(TN + FP)(TN + FN)}} \quad (1)$$

In detail, we calculated MCC using correct high-risk and low-risk predictions as true positives (TP) and true negatives (TN), respectively, while incorrect high-risk and low-risk predictions were used as false positives (FP) and false negatives (FN), respectively. For MCC, a value of −1, 0, and +1 correspond to an exact negative correlation, random prediction, and an exact positive correlation; We used precision to evaluate the proportion of truly high-risk patients from all the predicted high-risk patients; Recall to evaluate the proportion of predicted high-risk patients from the truly high-risk population; The precision and recall curve (PR-AUC) to assess performance at multiple decision boundary thredhsolds[29]. Whereas the MCC evaluates predictor performance for discrete outputs such as high-risk (1) and low-risk (0) at a fixed threshold, the PR-AUC evaluates the ranking ability of the algorithm. For instance, CLL-TIM outputs a continuous value ranging from 0 to 1, where anything above 0.5 is considered as a high-risk prediction and below zero considered as a low-risk prediction. The PR-AUC calculates the precision and recall of the algorithm at all possible thresholds from the continuous values generated by CLL-TIM. When used for the CLL-IPI[11], the precision and recall were calculated for each possible CLL-IPI score of 0–10 points.

This renders comparison of two algorithms without any bias that is introduced when committing to a single threshold.

For the above metrics, we generated confidence intervals (CIs) using 5,000 bootstrap sampled datasets of the internal test cohort and the CLL7 external cohort[41,66]. When algorithm variations generated predictions on different subsets from these cohorts, a new set of 5,000 bootstraps were generated. Sampling was performed with replacement and stratified to preserve original high-risk to low-risk patient ratios within each bootstrapped dataset. We then generated CIs of 95% for all metrics over the bootstrapped datasets. We performed comparison of CLL-TIM to CLL-IPI score models using one-tailed Wilcoxon signed-rank test on the difference in means of MCC and PR-AUC over the bootstrapped datasets. For comparison of models, which predicted on different subsets of the cohort like CLL-TIM High-Confidence (HC) and CLL-IPI NIH_7+/ CLL-IPI NI_4+, we performed a one-tailed Mann–Whitney U-Test on the difference in means of MCC and PR-AUC over the subsets of the bootstrapped datasets. As mentioned, we also generated ensemble models with different protocols than CLL-TIM (ENS-COMP$_{DC}$, ENS-COMP$_{3m}$, ENS-COMP$_{CLL-IPI}$). For comparison of the protocol that generated CLL-TIM (ENS-COMP) to the different protocols used as benchmarks (ENS-COMP$_{DC}$, ENS-COMP$_{3m}$, ENS-COMP$_{CLL-IPI}$) we used a one-tailed Wilcoxon signed rank test on the difference in 29 mean MCCs. We generated 29 ensembles (Size 2–30 base-learners) for each protocol, and pairing for the Wilcoxon signed rank test was performed with ensemble size.

**Clinical trial requirements.** In parallel with development of CLL-TIM, we aimed at implementing CLL-TIM for the selection of patients for a clinical phase II trial of preemptive BTK and BCL2 inhibition for newly diagnosed patients with CLL at high-risk of infection or treatment (PreVent-ACaLL, NCT03868722). To ensure a clinically meaningful hypothesis to be tested in the trial, the following requirements for the CLL-TIM algorithm were predefined:

  (i)  High-risk patients should have at least a 65% 2-year risk of infection or CLL treatment, and only patients predicted with high-confidence to be high-risk should be eligible for enrollment. At least 20% of patients should fulfilling these criteria.

  (ii) High-confidence predictions on at least 50% of the patients in the Danish nation-wide CLL registry should be available.

The high-confidence requirement was designed such as to minimize the risk of initiating the preemptive therapy in low-risk patients which CLL-TIM mislabels as high-risk. To derive high-confidence predictions from CLL-TIM, we used CLL-TIM's probabilistic risk (PR) through a process called soft-voting[53]. Using soft-voting in CLL-TIM, a patient was predicted as high-risk if the average PR of CLL-TIM's base-learners was greater than 0.5. This averaging was used as an estimate of predictive uncertainty (or predictive confidence). The rationale being that if the average PR was close 0 or 1 (i.e. not close to 0.5) then most of the base-learners in CLL-TIM agreed on a low- or high-risk prediction, respectively, and accordingly treated as high-confidence. Given this, the next step was thus to derive thresholds for what is considered as a high-confidence high-risk prediction, an uncertain prediction and a high-confidence low-risk prediction. For extracting the lower-bound threshold for a high-confidence high-risk prediction, we gradually reduced CLL-TIM's average PR from 1 until producing 20% high-risk predictions (criteria i) on the validation cohort in accordance with our clinical trial requirement. For high-confidence low-risk prediction, we similarly increased CLL-TIM's average PR threshold from 0 until producing 30% low-risk predictions (resulting in high-confidence predictions on 50% of the cohort i.e. criteria ii). The patients, for whom CLL-TIM's probabilistic output did not satisfy the thresholds for a high-confidence prediction, were considered as low-confidence/uncertain predictions. Using these requirements, this resulted in a lower-bound (average PR > 0.58) and upper-bound threshold (average PR < 0.28) for a high-confidence high-risk and high-confidence low-risk prediction.

**Risk factors.** To rank features according to their importance in the model prediction, we used SHapley Additive exPlanations (SHAP). SHAP is a unified framework for explaining the output of any machine learning model[31,56]. For the post-learning stage, it was important to understand how the features contributed to the predictive performance of CLL-TIM. Therefore, we were limited to using the in-built feature importance of each base-learner in CLL-TIM, such as XGB Feature Selection (FS), RF-FS etc. These methods do not satisfy the feature attribution property of consistency, i.e. a model may rely more than another model on a given feature, but it may still assign a lower importance[31,67]. Additionally, we needed a method that provided a consistent metric enabling us to generate the average feature importance values to create a single ranking for the ensemble. For the derivation of personalized risk factors, we needed a ranking method that could provide local/case-wise feature importance's for each patient and each base-learner. SHAP satisfies the feature attribution property of consistency[31,67], provides a single metric for feature importance across different learning algorithms and provides local interpretability for derivation of personalized risk factors[31,67]. An alternative to using SHAP is LIME[68], as it satisfies most of our requirements but the consistency property of feature attribution that SHAP guarantees, may be violated in certain instances with LIME[31]. For any given feature, the SHAP value (averaged across all of CLL-TIM's base-learners) quantifies the contribution of that feature towards predicting a patient as high or low risk. Negative SHAP values indicate

that the feature pushed a patient towards low-risk and positive values towards high-risk. SHAP values for the same feature may differ across patients. Thus for each patient, we ranked the top low-risk and high-risk features (according to their patient-specific SHAP value), thus representing personalized risk factors. For this, we averaged the SHAP value for each feature across all the base-learners in CLL-TIM and then displayed the top five risk factors pushing the patient towards low-risk and those pushing the patients towards high-risk. For population risk factors, the SHAP values were averaged across all patients in the training, validation and test cohorts. "Kernel SHAP explainer" was used for Logistic Regression and Elastic Network base-learners. "Tree SHAP explainer" was used for the Extra Trees, Random Forest and XGBoost tree-based models. We also perform univariate tests for all 7288 features and CLL-TIM's features using Student's $t$-Test, ANOVA $F$-Value, Mann–Whitney U-Test and Kruskal–Wallis tests. This enables us to confirm the risk factors discovered by CLL-TIM and SHAP, for their sole predictive power discriminating between high-risk and low-risk patients.

**Reporting summary**. Further information on research design is available in the Nature Research Reporting Summary linked to this article.

## Data availability
The individual patient level data that support the findings of this study are available from the corresponding author upon reasonable request. As the individual patient level data cannot be anonymized, only pseudonymized, according to Danish and EU legislation, the data cannot be deposited in a public repository. However, the authors provide a data repository with individual patient level data that can be made available with 2-factor authentication for researchers on request. The remaining data is available in the article and supplementary information files.

## Code availability
Code for CLL-TIM feature encoding, base-learner prediction and generation of personalized risk-factors is available at https://github.com/RA19/clltim.

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

## Acknowledgements

This work is in part supported by funding from the Novo Nordisk Foundation, grant NNF16OC0019302 and the Danish National Research Foundation grant 126. This work was initiated during a Modeling Immune System and Pathogen Camp (MispCamp). The authors thank the Danish hematology centers that participated with data submission to the Danish National CLL Registry. The following physicians contributed to data collection and represent the Danish Hematology centers participating in the Danish National CLL Registry: Christian Hartmann Geisler, Lisbeth Enggaard, Christian Bjørn Poulsen, Peter de Nully Brown, Henrik Frederiksen, Olav Jonas Bergmann, Elisa Jacobsen Pulczynski, Robert Schou Pedersen, Ilse Christiansen, and Linda Højberg Nielsen.

## Author contributions

R.A., J.La., and C.N. contributed to study conception, C.N. generated the original idea of the study, M.H., C.H. and J.B. provided data for the external validation, C.A. provided laboratory data for patients prior to diagnosis of CLL, M.A. and C.B. contributed to data cleaning and coding, R.A. developed the algorithm based on ideas and input from A.P., M.A., B.L., A.C., Y.L., J.B., J.S., M.J., C.M., M.T., M.F., J.La., and J.Lu. R.A. and C.N. wrote the paper, all authors commented the paper during the writing process and approved the final version. J.La. passed away during the preparation of this work. He contributed significantly to the work, however he did not have the chance to review the paper.

## Competing interests

C.N. has received consultancy fees and/or travel grants from Janssen, Abbvie, Novartis, Roche, Gilead, Sunesis, Acerta, AstraZeneca, and CSL Behring, research support from Abbvie, AstraZeneca, and Janssen, outside this project. C.D.H. received research funding and travel support from Roche. The other authors declare no competing interests.
