## [Peer Review File · Nature Communications]

Reviewers' comments:

Reviewer #1 (Remarks to the Author):

Major comments:

1. Why was the time-point 0 shifted to 3 months post-diagnosis when the whole evaluation is using methods for incorporation of missing data?
2. Discussion of selection bias is lacking.
3. Major issue :The use of any blood culture as a proxy for infection is questionable, at least in the way it is presented here. The citation from which it seems to get evidence is not published nor replicated while other longitudinal studies found frequent contamination in blood cultures in CLL (Kjellander et al. Ann Hematol 2016). This needs to be discussed more profoundly or maybe even completely revised.
4. Bag of words has two major issues which are not discussed: a) it has the curse of dimensionality issue as the total dimension is the vocabulary size. It can easily over-fit your model. The limited remedy is to use some well-known dimensionality reduction technique to your input data.
b) Bag of words representation doesn't consider the semantic relation between words. Generally, the neighbor words in a sentence should be useful for predicting your target word.
5. Please explain what the evidence and reasoning is to use uncommon words for a rare category as a proxy to comorbidity.
6. How did the authors ensure that distinct words really pointed to identical findings if the approach was completely data-driven?
7. The conclusion that a data-driven approach is more important for selecting variables than functional hypotheses is not supported by the argument before regarding immunoglobulin levels etc.
8. The authors' argumentation should be refined since it seems that the conclusion is already set in stones.
9. The practical aspect should be elaborated more in depth: What does it practically mean when I receive the report with overall low confidence after using the cll-tim.org information?
10. Despite moderate improvement in PR-AUC vs IPI, results remain modest while confidence intervals are overlapping, and results for the external german cohort are quite modest and differences are smaller. These limitations in practical relevance need to be critically discussed."

Reviewer #2 (Remarks to the Author):

The paper describes methods and results regarding the classification and the feature ranking of patients diagnosed with chronic lymphocytic leukemia from a Danish discovery cohort and a German validation cohort.

They name their tool CLL-TIM.

The authors applied several machine ensemble learning methods to classify the patients, by using correct data mining best practices:

- 1- They correctly split the dataset into training set, validation set, and test set;
- 2- They measured the performance of their binary predictions through several metrics, such as MCC, precision-recall curves, ROC curves, accuracy;
- 3- They used several classifiers from different machine learning families;
- 4- They used a discovery cohort and a separate independent validation cohort.

Regarding feature ranking, however, some improvements are needed:

5- The authors say they used the SHAP framework. Why did they not use typical machine learning methods for feature ranking, such as Random Forests, Support Vector Machine with radial Gaussian kernel, One Rule?

6- Relying completely on one method (SHAP, in this case), is never a good idea. The authors should use multiple methods.

7- For the risk factors identification, the authors should also use some traditional univariate biostatistics techniques, including some like Pearson correlation coefficient, Kruskal-Wallis test, Whitney-Mann U test, Wilcoxon signed-rank test, Student's t-test, chi squared test.

8- tSNE is an useful dimensionality reduction visualization technique, but it cannot replace traditional machine learning feature ranking methods. Also, the authors should include the results obtained through UMAP (<https://arxiv.org/abs/1802.03426>).

Other flaws:

9 - The authors say their tool handles efficiently missing data, but they do not explain how.

10 - The authors claim they predict "infection or CLL treatment", but this expression is confusing. What does it mean? Is an infection the same thing of a CLL treatment? Are they predicting both? Why are these two concepts treated in the same way?

Style flaws:

11 - The grammar is good but all the passive verb forms must be replaced with active verb forms.

Response to Reviewer #1

We thank you sincerely for your thorough review of this manuscript. Your comments have alerted us to areas which will greatly improve the final paper. In addition to several amendments to the manuscript text, we now provide a detailed justification for blood cultures as a proxy for infection and an additional analysis on positive blood cultures. Additions to the manuscript relative to your review include two new main figures (Figure 4, 5d and 5e) and three new pieces of supplementary data (Supplementary Figure 2, 10 and Extended Data Figure 2). The manuscript and supplementary index have the changes and additions highlighted in red. We hope you find the revised manuscript satisfactory.

1. Why was the time-point 0 shifted to 3 months post-diagnosis when the whole evaluation is using methods for incorporation of missing data?

We thank you for the opportunity to further clarify this. While we shift time-point 0 to a maximum of 3 months post-diagnosis during the training phase to allow for all diagnostic tests to be available, there is no restriction to have the time-point 0 specifically at 3 months post diagnosis for the prediction phase. In the internal test cohort, depending on when the last test was available, time-point 0 for each patient was distributed at various points in the 0 to 3 months' time-window. For the external German cohort, time-point 0 for patients was distributed along 0 to 1 year post-diagnosis (See new Extended Data Figure 2). While we are primarily interested in making predictions close to CLL diagnoses, CLL-TIM may be used at any time-point post CLL diagnosis. Even though CLL-TIM works with missing data, for the training phase, we wanted to minimize the missing data as much as possible. For example, IGHV status, FISH and TP53 mutations in most centers take between one to three months for the results to be available, and we therefore wanted to make sure to have as much of this information available when training CLL-TIM. Also, some patients (n=373 in our Internal Cohort, Extended Data Fig.1) will need treatment within the first 3 months post-diagnosis. The time-window of 3 months allows us to remove these patients from analysis, as these patients are already identified by the clinicians as potentially needing treatment at time of diagnosis. Therefore, including these patients, and predicting them correctly, would have positively biased our results.

To clarify this matter, we have added additional text in:

'Results - Patient Characteristics from the Danish National CLL registry' – Page 4.

We thus excluded patients who died (n=74) or initiated CLL treatment (n=373) prior to this (Extended Data Fig.1), reducing the available sample size to n=3,720 (see Table 1 for baseline characteristics). For assessment in the internal test cohort, time-point zero varied between zero to three months post-diagnosis and for the external test cohort it varied between zero to one-year post-diagnosis (Extended Data Fig. 2)

2. Discussion of selection bias is lacking

Thank you for pointing out that we need to address this further. We have now added the following in:

Discussion – Page 9

CLL-TIM predicted infection and CLL treatment with similar frequencies as first events, thus validating the ability of CLL-TIM to identify both patients at risk of infection and in need of CLL treatment. Predictions on the entire test cohort (n=646) are representative of the Danish CLL population as CLL-TIM is modeled upon a registry with 98% coverage of patients diagnosed with CLL in Denmark³⁸. However, the Danish

cohort represents a more indolent population than in most other published cohorts^{11,38}. Patients with incomplete diagnostic work up fare worse than patients with full diagnostic work up³⁹. This may affect predictions on the subset of patients with full CLL-IPI (BENCH-I, n=288 and BENCH-E, n=281 cohorts).

Results – CLL-TIM is Robust to Missing Values on an External German Text Cohort, CLL7

This resulted in an average missing feature rate ranging from 42% to 58% per base-learner; while the cohort overall had indolent baseline characteristics with half of the high-risk patients randomized for early treatment and thus removed prior to this analysis (Table 1).

Discussion – Page 11

For the external German CLL7 validation cohort, we observed a lower call-rate of high-risk predictions both by CLL-IPI and CLL-TIM. Partly explaining this, is that half of the high-risk patients in the external German CLL7 validation cohort were randomized for early treatment and not part of our predictions.

2. Major issue: The use of any blood culture as a proxy for infection is questionable, at least in the way it is presented here. The citation from which it seems to get evidence is not published nor replicated while other longitudinal studies found frequent contamination in blood cultures in CLL (Kjellander et al. Ann Hematol 2016). This needs to be discussed more profoundly or maybe even completely revised

We thank you for your comment and our apologies for not including the proper citation in the manuscript. We agree that this definition needs further clarification and justification in the manuscript. We use the event of having a blood culture drawn (independent of the result of the blood culture) as a proxy for clinical infection in our registry. This is justified, as according to Danish guidelines and as standard practice, when a patient exhibits symptoms of a serious infection, a blood culture is drawn and not awaiting the result, the patient is treated with antibiotics (while the type of antibiotics may be subsequently changed based upon results of the blood culture). Thus, this is considered as a clinical infection. While blood cultures drawn from patients diagnosed with CLL are more often positive than blood cultures drawn from the background population¹, the event of a blood culture with contamination or a negative blood culture does not mean that the patients did not have a clinical infection, but that the infectious agent could not be identified by the current diagnostic set up²⁻⁴. The drawing of a blood culture as a proxy for clinical infection, has been previously used by us and others^{1,5,6} and an increased risk of mortality for CLL patients with infections in the first year of diagnosis, has been found both when using a blood culture as a proxy for infection⁶ and when only using infections annotated as serious by physicians^{7,8}. For example, in our previous work we found that patients with infections prior to CLL treatment have a higher 30-day mortality (9.8%) than upon CLL treatment, leading to worse treatment-free survival and OS compared to matched patients without severe infection^{7,9}. As also validated by others^{7,9}. Given that only 10% of blood cultures in the CLL population are detected as positive, if we were to restrict the definition of an infection to that of a blood culture with a positive finding, we would miss 90% of the patients with a clinical infection, who would be annotated as low-risk in our target outcome. Subsequently, future patients with similar characteristics, would also go undetected by models like CLL-TIM, thereby excluding them from consideration in pre-emptive treatment clinical trials like the PreVent-ACaLL. We believe that some insights into the

informative content of our definition for infection may also be gathered from our data-driven results. For instance, both our infection features and infection outcomes were annotated using the drawing of blood culture as a proxy for infection. If the drawing of a blood culture severely underestimated truly positive infections, or did not represent any clinically relevant event, we would expect the infection features to not have any informative content to be predictive of infection nor of treatment. Similarly, we would see no significant differences in the performance between models predicting the infection outcome, as this outcome would have no learnable information. Both of these expectations are contrary to what we see in our work, as infection features are amongst the top prognostic factors both when tested for their sole prognostic power (Supplementary Table 6) and in multivariable models (Fig. 6b and Supplementary Table 1). Furthermore, we observe significant differences between models in the predictions for patients who have infections but no treatment (Fig 5). To gather some insight on how predictions are affected when using other infection definitions, we compared test performance on 1+ blood cultures (our original definition) against, 2+, 4+ and positive finding blood cultures. Recall was higher for positive findings in high-confidence predictions (Fig. 5d) but this increase was not found to be significant, at least at this sample size. Recall rates were very consistent when not restricting to high-confidence predictions (Fig. 5e). The fact that we see significant differences in performance between CLL-TIM and CLL-IPI, for 1+ blood culture, suggests that there is learnable information under this definition. While we cannot rule out that for some infections, we may be annotating events that have similar symptoms to infection but do not represent true infections, from our work we can see that using the drawing of a blood culture as a proxy for clinical infection conditioned on it being standard practice to do so after suspicion of an infection, holds both prognostic and learnable information.

We have amended the manuscript accordingly:

Results - Patient Characteristics from the Danish National Registry - Page 4:

For modelling, we only used patient data prior to the prediction point. As a composite outcome, we set out to predict the combined event of an infection or CLL treatment within 2-years from the prediction point. As it is standard practice that a blood culture is drawn when a patient has symptoms classified as a serious clinical infection, as a proxy for infection, we used the event of having a blood culture drawn. This is irrespective of whether the blood culture was negative, indicative of contamination or positive²⁶⁻²⁸.

Results - Immune Dysfunction is Linked to Progressive Disease - Page 7

We next assessed how the recall for infections, differs when changing the definition for this outcome (Fig. 5). For high-confidence predictions, recall increased for patients with positive or multiple blood cultures. However, none of these increases were significant ($p>0.05$). In turn, for all outcomes, CLL-TIM achieved significantly higher recalls than CLL-IPI.

Discussion - Page 10:

In this work we find that using the drawing of a blood culture as a proxy for serious clinical infection, in agreement with guidelines and standard practice to draw a blood culture upon suspicion of a serious infection, holds both prognostic (Fig. 6b and Supplementary Tables 1 and 6) and learnable information (Fig. 5 and Supplementary Table 6). We cannot rule out however, that for some infectious events, we may be annotating events that have similar symptoms to infection but do not represent true infections, as is also the case in the clinical setting. The increased risk of mortality for CLL patients with infections in the first

year of diagnosis has been demonstrated both when using a blood culture as a proxy for infection³⁵ and when using infections classified as serious by physicians^{7,41}. Though we achieved a 100% recall for positive blood cultures in CLL-TIM's high-confidence predictions, the patient count was too low for any significant differences to be observed. Namely, we found no significant differences in the recall for outcomes of one or more blood cultures and blood cultures with positive findings.

4. Bag of words has two major issues which are not discussed: a) it has the curse of dimensionality issue as the total dimension is the vocabulary size. It can easily over-fit your model. The limited remedy is to use some well-known dimensionality reduction technique to your input data.

Thank you for allowing us to address and clarify these issues. We agree that the resultant feature size from BOW has a very high dimension. We therefore reduced the dimension of BOW prior to training in two ways:

- i) The majority of the high-dimension is due to the long tail created by the rare words which only occur in less than 1% of the CLL population. From 1572 pathology words, 1419 (90.2%) were rare, and from 3386 diagnosis words, 3170 (93.6%) were rare. For these rare words, we represented them with one variable that increases its value by 1, each time a rare word is found for a patient. Thus, the pathology and diagnosis BOW vector (dimension) size was reduced to 153 and 216, respectively.
- ii) Additionally, after appending the BOW vector to the feature vectors from the other feature generation methods (Fig. 1c), we applied several dimensionality reduction techniques before training the base-learners (Fig. 1d, Methods 'Base-Learner Generation' and Extended Data Table 5). To reduce the bias introduced by a single dimensionality reduction technique, we used four different methods (XGBoost Feature Selection (FS), Random Forest FS, ExtraTrees FS, and ANOVA F-Value FS). As a result, no base-learner was trained on more than 150 features, thereby reducing the risk of over-fitting, whilst also keeping a diverse pool of reduced features for a more diverse ensemble.

To emphasize that we applied dimensionality reduction, we added the following text in:

Results - Development and Composition of CLL-TIM – Page 4:

This resulted in a final feature space of 7,288 dimensions (Extended Data Table 2), reduced using dimensionality reduction techniques (Fig. 1d, Methods 'Base-Learner Generation' and Extended Data Table 5), upon which we applied 2,000 different algorithms (referred to as base-learners).

4b) Bag of words representation doesn't consider the semantic relation between words. Generally, the neighbor words in a sentence should be useful for predicting your target word.

We agree that Bag-of-Words (BOW) has a clear disadvantage when used for documents or images, as the neighboring relationship between words in a sentence is not modelled in the feature space. In our case however, proximity or spatial relationships do not exist between words, as words were not retrieved from documents of plain text. Rather, we used pathological diagnostic coding and microbiology findings directly from their sources of origin. Therefore, we believe BOW has no drawbacks in this sense as there are no neighbors to exploit. What does exist however is semantic or functional relationships (i.e. variables having

similar information with respect to target variable but represented by different words/codings) between words. As you highlight, BOW in its original form does not account for that in its feature space. At worst, what this means is that we are not indicating *a priori* to the learning algorithm, that two words have a semantic or functional relationship. This however does not exclude the learner from modelling such a relationship during the learning phase. Let's say two words are semantically similar, and functionally also similar, then, a tree-based learner (such as Random Forest (RF), XGBoost and Extra Trees (ET)) are able to learn 'OR' functions for these variables and others alike. If we wanted to include any *a priori* knowledge on semantic or functional relationships, the two methods that we believe can address this significantly also come with significant caveats that restrict us from implementing them currently:

- i) *Standardizing the semantics and a priori assigning identical findings to the same word:* The challenge for this approach if at all possible in its entirety due to the combinatorial nature of this problem, is that we would be imposing our own bias on what we think is related or not, thus jeopardizing the whole data driven approach, and potentially hindering the learning process rather than helping it.
- ii) *Not employing any standardizing, and having a deep-learning model where you represent each and every word using embeddings:* Using embeddings, a deep-learner could automatically learn associations by itself and put such words (with semantic or functional similarity) close together in the dimensional space that is restricted for embeddings¹⁰. The caveat to this is that learning these semantic and functional relationships between words as described, while is conducive of a fully automated approach, necessitates several-fold more patients than we currently have access to. Using the current work as a proof-of-concept of a data-driven approach for CLL prognostication, we hope that in the future we would be able to access data for more patients, so that such an approach becomes implementable.

Given a number of the comments are related to BOW, we would like to clarify that BOW is only one of the methods used to generate features, amongst several others (Fig. 1c). After rounds of automated feature selection and learning, BOW features formed a minor part of the final CLL-TIM model. From the top 100 ranked features in CLL-TIM, only eight were BOW (Supplementary Table 1). Additional analysis on the effect of adding pathology and diagnosis data from BOW resulted in minimal changes to CLL-TIM's probabilistic output (Supplementary Figure 10). With this in mind, we cannot ascertain whether BOW features were not critical for predictions because of the rudimentary way of representing categorical data or because there is no relevant information within them for prediction of treatment and infection.

We have amended Discussion - Page 12 (see bold):

BOW features that modelled pathology and diagnosis were selected by our data-driven strategy, but had small effects on performance (Supplementary Figs. 6 and 10). Patients with a high number of rare pathologies showed only a minor increase in risk of infection and treatment (Supplementary Fig. 16). It cannot be ascertained to which extent this is due to comorbidity or other events modelled by this feature. Access to larger patient cohorts in the future may enable pathology and diagnosis information to be modelled with word embeddings that may better exploit relationships between these types of categorical features⁵⁵.

5. Please explain what the evidence and reasoning is to use uncommon words for a rare category as a proxy to comorbidity.

Thank you for highlighting this and allowing us to further justify this interpretation. First, we would like to clarify that the sentence regarding proxy to comorbidity of uncommon words (Methods ‘Feature Generation’) refers to pathology and diagnosis variables and not urine cultures/blood cultures. We have clarified this in the manuscript (see below). For the high number of rare pathologies/diagnoses (those found in less than 1% of CLL population), we hypothesized that comorbidity is likely to be represented in part by these, which may or may not have an effect on risk of treatment and infection. Upon creating the rare categories (1419 rare words for pathology, and 3170 rare words for diagnoses), our physicians went through a random selection from the list and confirmed the presence of several comorbidity related events. These however also included non-comorbidity related events and several words that point to similar findings for the same patient. Therefore, the rare diagnosis and rare pathologies can be thought of a proxy to comorbidity, but with additional noise of double counting and non-comorbidity related events. Even though the actual number of rare words may not directly represent the rate of comorbidity (given the additional noise), in its extremities, patients with a high rare word count we hypothesize are more likely have a higher number of comorbidity related events than those with a low rare word count. In this scenario, non-linear methods like the tree-based methods used in this work, still have the capacity to extract relevant information from such variables. Prompted by your question we performed additional analysis to understand further how CLL-TIM uses these variables.

The count of rare diagnoses and rare pathologies were included in the final CLL-TIM model based on the solely data-driven approach, suggesting that the information in these variables is useful for predicting risk of infection and treatment. ‘Rare Diagnosis’ was used by less than 10% of the base-learners (Supp. Table 1), however 40% of base-learners used ‘Rare Pathology’ as a feature (Fig. 2a), suggesting that this variable is indeed more informative. Given that the tree-based learners (Random Forest (RF) and Extra Trees (ET)) made use of ‘Rare Pathology’ (Fig. 2a), and not linear methods like Linear Regression (LR) and Elastic Network (EN), is in line with the reasoning that the number of rare pathologies is a noisy representation of comorbidity, but can be informative when using thresholds (like in RF and ET) instead of raw counts (like in LR and EN). Additional analyses of the rare pathology feature for the CLL population shows that an increase in number of rare pathologies is positively correlated to the risk of infection and treatment (**Supplementary Fig. 16**). Specifically, above circa 4 uncommon words, as the number of uncommon words increases, there is a continuous increase in risk of infection/treatment. Interestingly, CLL-TIM has also learned to limit the contribution of ‘Rare Pathology’ when the number of words is higher than circa. 20, thereby not overestimating the risk contribution in these cases. It may be that at these counts, non-comorbidity related events and identical findings increase at a higher rate than the comorbidity related events and thus risk may be overestimated. Therefore CLL-TIM has learned to saturate the contribution for high counts. It should also be noted that the range of contribution of ‘Rare Pathology’ to CLL-TIM’s probabilistic output (-0.005 to 0.015) is minor compared to the risk factors presented in Fig 6d.

We have thus added our pre-learning phase reasoning for using rare pathology/diagnoses as a proxy for comorbidity in the **Online Methods** section ‘**Feature Generation**’ and have amended the **Discussion** with our findings related to these features.

Online Methods – Feature Generation - Page 15:

Based on medical review of the highest ranked uncommon words in the rare categories for pathology and diagnoses, we hypothesized that these could be used as a proxy for comorbidity. Even though the actual number of rare words may not directly represent the rate of comorbidity (given the additional noise of non-

comorbidity related words and different words pointing to the same finding), patients with a high rare word count were hypothesized to be more likely to have a higher number of comorbidity related events than those with a low rare word count. Using the BOW paradigm, each patient was thus represented using a vector with the length of all the combined vocabularies (i.e. 424 words).

Discussion – Page 12 (See Bold):

BOW features that modelled pathology and diagnosis were selected by our data-driven strategy, but had small effects on performance (Supplementary Figs. 6 and 10). Patients with a high number of rare pathologies showed only a minor increase in risk of infection and treatment (Supplementary Fig. 16). It cannot be ascertained to which extent this is due to comorbidity or other events modelled by this feature. Access to larger patient cohorts in the future may enable pathology and diagnosis information to be modelled with word embeddings that may better exploit relationships between these types of categorical features⁵⁵.

6. How did the authors ensure that distinct words really pointed to identical findings if the approach was completely data-driven?

During the data cleaning process, we removed any highly correlated variables with Pearson's Correlation Coefficient (PCC) >0.98 (Extended Data Table 2) and therefore if two distinct words are highly correlated one of them is dropped. Prompted by this question we have performed a correlation analysis between CLL-TIM's 228 features and present the results in **Supplementary Fig. 2**. With only 2% of all possible pair-wise feature correlations having a $|PCC|>0.8$ and with 75% of all possible pair-wise feature correlations having $|PCC|<0.2$, we observe a low feature redundancy in the 228 features in CLL-TIM. Whereas redundancy would not affect the performance of learners like XGB, RF and ET, for highly correlated features, it may affect feature ranking as feature importance would be shared between these features thereby underestimating their importance.

We have included our findings on feature redundancy in:

Online Methods – Feature Generation - Page 15:

Due to differences in nomenclature, distinct words pointing to identical findings were in fact left untouched, as we strived for a fully data-driven approach without interference. Any words with high correlation were however removed as part of the data-cleaning process (Extended Data Table 2).

Results' - 'Development and Composition of CLL-TIM - Page 5:

In total, CLL-TIM uses 85 original variables from patient histories (Fig. 2a), which translate to 228 engineered features (Fig. 2b and Supplementary Table 1). CLL-TIM also exhibited low redundancy among the selected features, where only 2% of all possible pair-wise feature correlations had an absolute Pearson's Correlation Coefficient (PCC) greater than 0.8 (Supplementary Fig. 2).

Supplementary Fig. 2:

The highest correlations were observed between the date of tests for eosophilocytes, basophilocytes, lymphocytes, neutrophil vount, leukocytes and hemoglobin; the dates of tests for Blast Cells, Myelocytes and Metamyelocytes; the values for leukocytes and lymphocytes; as expected due to these variables being included in the same medical order (differential count). Similarly, for missing value indicators of the different FISH status variables, a high correlation was seen. For these features, feature importance may

be underestimated in our model. However, with 75% of all possible pair-wise feature correlations having $|PCC| < 0.2$, we expect the underestimation of feature importance to be minimal. The low feature redundancy exhibited in CLL-TIM suggests that it uses complimentary information in features for its predictions.

7. The conclusion that a data-driven approach is more important for selecting variables than functional hypotheses is not supported by the argument before regarding immunoglobulin levels etc

We thank you for the option to rephrase this, as we realize that this argument was not clearly put forward. When testing for the significance of a variable (say the role of Immunoglobulin's (Ig) in risk of infection), where we only test within a pool consisting of small number of variables, we may easily overestimate the importance of any variable. In contrast to previous work on immunoglobulins^{9,11,14,1}, we find no evidence that they play any significant role in predicting either treatment or infection. While immunoglobulins show univariate significance, for composite and infection outcomes (Supplementary Tables 5 and 6), they are markedly outranked by other features.

We have re-worded this section of the discussion and included additional supplementary (Supplementary Tables 5 and 6) to explain ourselves better:

Discussion - Page 11:

In contrast to models that simply add the impact of individual risk factors^{9-11,20-23}, we uncovered the more complex and non-linear mechanisms by which prognostic factors contribute toward composite risk in CLL – this, both on a population (Fig. 6d and Supplementary Table 6) and individual level (Fig. 7b and c). None of CLL-TIM's base-learners made use of immunoglobulins and 'Doctor's Choice' models that included immunoglobulins, were outperformed (Supplementary Fig. 5). Immunoglobulins were also substantially outranked by many other features in our univariate analyses (Supplementary Tables 5-6). These results are in contrast to previous work on immunoglobulins^{9,11,14,1}. It is possible, that their importance was previously overestimated as they were only benchmarked against a handful of variables.

8. The authors' argumentation should be refined since it seems that the conclusion is already set in stones.

Thank you for pointing this out. We have changed accordingly throughout the manuscript:

Page 3: Finally, the PreVent-ACaLL trial will seek to further validate CLL-TIM in clinical use, ~~thus paving the road for~~ and serve as an example of explainable machine learning for personalized treatment in CLL.

Page 9: A first step in ~~changing the~~ attempting to change the ramifications of immune dysfunction in CLL is to identify high-risk patients prior to any infection or CLL treatment¹¹

Page 12: As infections remain the major cause of death in CLL⁹, with increased incidence after chemoimmunotherapy¹¹⁻¹³, identification of patients at risk of infection and the provision of personalized risk factors ~~mark a major leap towards~~ can aid efforts aimed at personalized treatment for CLL.

Page 12: ~~With the aim of changing the natural history of immune dysfunction and infection-related morbidity and mortality for patients with CLL, the~~ .-The next step is using CLL-TIM for patient selection

in the investigator-initiated, randomized phase 2-3 trial within the Nordic and the Hovon CLL study groups, the PreVent-ACaLL trial (clinicaltrials.gov: NCT03868722).

Page 13: Addressing the lack of prospective validation for machine learning in the clinic¹¹, through the PreVent-ACaLL trial we will be able to ~~prospectively validate~~ **further assess** the predictive performance of CLL-TIM, as the observation arm and the patients predicted with low risk or low confidence predictions will be followed for infections and CLL treatment.

Page 13 By deriving personalized risk factors, ~~we are opening the black box of machine learning thereby paving the road towards~~ **we can make machine learning more explainable while providing a resource for efforts aimed at** delivering personalized medicine.

9. Theoretical aspect should be elaborated more in depth: What does it practically mean when I receive the report with overall low confidence after using the cll-tim.org information?

Thank you for allowing us to further elaborate on this important aspect of CLL-TIM. For this, we have added a new main figure and supplementary (Fig.4 and Supplementary Fig. 10), which shows the practical meaning for a physician when receiving a report with low- or high-confidence (Fig 4b), Additionally, how the addition of patient data affects confidence over the whole test set population (Fig. 4a) with specific patient examples (Fig. 4d) – thereby showing how confidence may be increased for low-confidence patients to make a prediction useful in the clinical setting.

We have amended the manuscript with:

Results – a new Section entitled **Addition of Patient Data Increases CLL-TIM's Performance and Confidence in Predictions** in Page 7:

... *The addition of patient data, also had the effect of increasing CLL-TIM's confidence in its predictions (Fig. 4a), thereby increasing the operating Hazard-Ratio of CLL-TIM (Fig 4b, Supplementary Fig. 3). Predictions using only baseline variables, were of mostly low-confidence. Infection and laboratory histories had the most marked effects in increasing CLL-TIM's confidence, whilst pathology and diagnoses variables had more subtle effects (Supplementary Fig 10). The addition of patient data from each variable category, affected the confidence for each patient differently (Fig. 4c).*

Discussion - Page 11:

In line with work deriving uncertainty estimates from ensembles^{51,52}, CLL-TIM's uncertainty estimates allows the treating physician to assess whether a predicted risk is actually of clinical significance for the patient. As exhibited by the low HR for low-confidence patients (Supplementary Figs. 3c and 3f), a low-confidence prediction means that for this patient, CLL-TIM cannot give a more accurate prediction. For a high-confidence prediction, CLL-TIM is more certain of its prediction and operates at a HR of 7-9 (Supplementary Figs. 3a and 3d). A related benefit of CLL-TIM, is that upon input of further patient data, low-confidence predictions may still be converted into high-confidence predictions (Fig. 4).

10. Despite moderate improvement in PR-AUC vs IPI, results remain modest while confidence intervals are overlapping, and results for the external German cohort are quite modest and differences are smaller. These limitations in practical relevance need to be critically discussed.

Thank you for allowing us to elaborate on the performance of CLL-TIM as compared to previous prognostic indices in CLL.

For CLL-TIM's performance on the German cohort, we expected the results to be similar to that of CLL-IPI. In this cohort, due to missingness of data (Supplementary Fig. 7), CLL-TIM has at its disposal, a limited number of variables other than those forming CLL-IPI. Nearly no data is available for past infections and prior medical/pathology history. CLL-TIM has therefore no means to arrive at any markedly different conclusions than CLL-IPI. We have shown how adding data prior to CLL diagnosis increases performance (Supplementary Fig. 9). In this revised manuscript, we also find that adding patient data prior to CLL diagnosis, increases confidence in predictions (Figure 4a and Supplementary Figure 10), and can help detect high-risk patients that would have been predicted as low-risk when using baseline variables (Figure 4a and Supplementary Fig. 10), or when using CLL-IPI (Fig. 4c – See P₀₁-P₀₃). What we also failed to mention in our initial submission, was that half of the very high-risk patients were excluded from the CLL7 study cohort (due to inclusion in the treatment arm of the study), and thus this cohort consisted of patients with more indolent baseline characteristics. The need for data other than baseline data, which is already pointing to indolence, is even more pertaining in these situations. With this in mind, performance on the German cohort was still significantly better than CLL-IPI ($p < 0.05$). We believe that the results on the German cohort are evidence to the importance of modelling infection history and patient data prior to CLL diagnosis. We would like to point out that even in situations with limited data, it is still advantageous to use CLL-TIM over CLL-IPI, firstly because CLL-TIM notifies you on how trustworthy that prediction is, and secondly because there is the opportunity to extract more patient data and get better predictions with CLL-TIM.

With regards to the confidence intervals (CIs), whereas for PR-AUC and high-confidence predictions there are minimal to no overlaps in CIs, when using MCC and when assessing predictions that also include low-confidence predictions, CIs do indeed overlap greatly, even though the difference is statistically significant. The reason why we see statistical significance even when CIs are highly overlapping (like when not restricting to high-confidence patients), is because for these patients, we could perform paired tests like the Wilcoxon signed-rank test. In these situations, looking at the CIs for two models can be misleading, as it is the distribution of the change in performance for each bootstrap on which the significance is calculated on. The source of the overlapping CIs. We believe arises mostly arises from how we generated the CIs. Whereas we can generate CIs automatically for measures like Accuracy, as each prediction is a Bernoulli trial with a binomial distribution and hence we can approximate with a Gaussian distribution for large samples¹¹, the MCC and PR-AUC cannot be approximated this way. Nor did we have multiple test sets for us to generate CIs from the results across different test sets. This limited us to use bootstrapping on a single test set to generate our CIs. Given that we have a low proportion of high-risk patients in our test sets, the change in the number of high-risk patients detected in each random bootstrap affects results like the MCC greatly (See Table 1 below). Even though the Accuracy ranges from 0.89-0.96, the MCCs ranges from 0.39 to 0.84.

Table 1. MCC Variability in Bootstrap

	Bootstrap Sample 1	Bootstrap Sample 2	Bootstrap Sample 3
Total Bootstrap Size	300	300	300
Number of High-Risk Patients in Bootstrap	40	40	40
Number of Low-Risk Patients in Bootstrap	260	260	260
Percentage of High-Risk Patients in Bootstrap	13%	13%	13%
True-Positives	36	9	30
False-Positives	8	2	20
True-Negatives	252	258	240
False-Negatives	4	31	10
Total	300	300	300
TP-Rate	0.900	0.225	0.750
PPV	0.818	0.818	0.600
MCC	0.835	0.393	0.614
Accuracy	0.960	0.890	0.900

Finally, we would like to point out, that the contribution of CLL-TIM relative to those of CLL-IPI are not limited to only performance. We have put great effort in designing an algorithm with clinical relevance that includes the generation of trustable confidence estimates, the ability to work under missing values and that of generating personalized risk factors for CLL patients. CLL-IPI has a low performance for detecting risk of infection as not developed for this purpose (Fig 5d and e, Supplementary Fig. 14), while infections are now the leading cause of mortality in patients with CLL. We hope that this work has also sufficiently put forward a case for the importance of data collection prior to CLL diagnosis for prognostication.

We have amended the **Results** and **Discussion** sections to address the limitations related to overlapping CIs and those related to the external German cohort.

Results - CLL-TIM is Robust to Missing Values on an External German Test Cohort, CLL7 – Page 6

This resulted in an average missing feature rate ranging from 42% to 58% per base-learner; while the cohort overall had indolent baseline characteristics with half of the high-risk patients randomized for early treatment and thus removed prior to this analysis (Table 1).

Discussion – Page 11:

For the external German CLL7 validation cohort, we observed a lower call-rate of high-risk predictions both by CLL-IPI and CLL-TIM. Partly explaining this, is that half of the high-risk patients in the external German CLL7 validation cohort were randomized for early treatment and not part of our predictions. Additionally, the external cohort has limited data besides baseline variables, which we have shown to have very limited separability of high-risk and low-risk patients (Fig. 4a). Further corroboration to this was found through simulations on the internal cohort, where certain high-risk patients could only be correctly identified with the addition of infectious history and laboratory variables. Similarly, upon performing simulations on our internal cohort that mimic the external cohort, both recall of high-risk predictions and detection of infections was reduced when using only post-diagnosis data.

Discussion - Page 9/10:

Bootstrapping was necessary for MCC and PR-AUC, as CIs for these metrics cannot be automatically generated by using the gaussian approximation³⁸. One limitation of comparing model performance on a single test set using the bootstrap method^{39,40}, is that for small sample sizes, CIs were sometimes wide and overlapping. Particularly, when low-confidence predictions were included. However, even in these conditions, CLL-TIM still achieved significant performance improvements over CLL-IPI.

Response to Reviewer #2

The paper describes methods and results regarding the classification and the feature ranking of patients diagnosed with chronic lymphocytic leukemia from a Danish discovery cohort and a German validation cohort.

They name their too CLL-TIM.

The authors applied several machine ensemble learning methods to classify the patients, by using correct data mining best practices:

- 1- They correctly split the dataset into training set, validation set, and test set;**
- 2- They measured the performance of their binary predictions through several metrics, such as MCC, precision-recall curves, ROC curves, accuracy;**
- 3- They used several classifiers from different machine learning families;**
- 4- They used a discovery cohort and a separate independent validation cohort.**

We thank you for reviewing our paper and your insightful suggestions have helped make our analysis on risk factors more thorough. In response to your comments, we have updated the manuscript with additional feature ranking methods (**Supplementary Tables 4-6**) and clustering of personalized risk-factors with UMAP (**Supplementary Fig. 15**). We have added a new methods section dedicated to the handling of missing data and provide further details on our justification for our post-analysis ranking methods. We have also re-modelled a main figure (previously 4b now **5a**) on the link between immune dysfunction and treatment. The manuscript has also been revised with active voice.

The manuscript and supplementary index have the changes and additions highlighted in red. We hope you find the revised manuscript satisfactory.

Regarding feature ranking, however, some improvements are needed:

5- The authors say they used the SHAP framework. Why did they not use typical machine learning methods for feature ranking, such as Random Forests, Support Vector Machine with radial Gaussian kernel, One Rule?

6- Relying completely on one method (SHAP, in this case), is never a good idea. The authors should use multiple methods.

We thank you for your comments and appreciate the opportunity to clarify further what feature ranking methods were used for pre-learning and post-learning stages and the motivations behind them. We also thought it would be best to address your two concerns in (5) and (6) simultaneously. SHAP provides a unified metric for different machine learning algorithms, however the rankings are still generated based on

the individual learning algorithms themselves, their hyper parameters, and their features. This means that when we use SHAP to rank features in CLL-TIM, we have effectively generated 28 different feature rankings that include rankings from five different types of learning algorithms (XGBoost (XGB), Random Forest (RF), Extra Trees (ET), Logistic Regression (LR) and Elastic Network (EN) reflecting how each base-learner is using its features. At the pre-learning stage, we aimed for diversity to improve robustness to missing data and thus we assembled 12 different feature rankings generated using four feature ranking methods that rank three outcomes each (infection, treatment, infection or treatment - See Methods ‘Base-learner generation’, Fig.1 d and Extended Data Table 5). In the post-learning stage, the choice of feature ranking was based on i) ranking accuracy (since we aim to interpret the rankings as risk factors), ii) the ability to unify rankings across different base-learners and iii) the ability to explain local predictions for subsequent derivation of patient personalized risk factors. The only methodology to our knowledge currently able to do this is SHAP.

In further detail:

For the pre-learning stage, we aimed to reduce the feature size (from 7288 down to a maximum of 150) but to also have within this reduced set, features which have high impact while being diverse. Therefore, for the pre-learning phase we ranked features using Random Forest Feature Selection (RF-FS), Extra Trees (ET-FS), XGBoost (XGB-FS), and for Elastic Networks (EN), Logistic Regression (LR) and Perceptron we used ANOVA F-Value FS (Fig.2d and Extended Data Table 5). As observed by CLL-TIM’s composition (Fig. 2a – see clustering of base-learners), different methodologies tended to include different variables. For example, the linear methods (EN and LR) uniquely opted for (Hematocrit, Albumin.... Metamyelocytes – see bottom right cluster in Fig. 2a). ET and RF avoided laboratory variables, uniquely chose ‘Rare Pathology’ and does not include blood cultures whilst XGB had a more mixed selection of features. This diversity would not have been possible without several feature ranking methods in the pre-learning phase.

A note on Support Vector Machines: SVMs (along with SVM-FS), both linear and radial basis kernels, were initially included among the learning algorithms and FS methods, as they could provide further diversity in the ensemble features. However, the available scikit-learn¹² implementations did not allow us to save the ‘random state’ that controls for the stochastic aspects of SVMs. This meant that multiple predictions on the same patient resulted in inconsistent probabilistic outputs. This was not an issue for the other learning algorithms with inborn stochastic aspects, as we were able to save the ‘random state’ as a parameter along with all the other hyper-parameters. We therefore had to exclude SVM from the final pool of base-learners and feature ranking methods.

For the post-learning stage it was important for us to understand how the features contributed to the predictive performance of CLL-TIM. Therefore, using any feature ranking method that ranks features independently of the implemented model, or independently from its interaction with other features, would not be suitable. Thus, we were restricted to use the in-built feature importance measures of each base-learner in CLL-TIM (so XGB-FS for the XGB base-learners and so forth for RF, ET, LR and EN). There are three issues with doing so:

- i) The tree-based learners XGB, RF and ET, which constitute 24 from 28 of our base-learners, use gain¹³, split count¹⁴ or permutation methods¹⁵ to calculate feature importance¹⁶. These methods do not satisfy the feature attribution property of consistency, i.e. a model may rely more than another model on a

given feature, but it may still assign a lower importance^{16,17}. We could do away with this in the pre-learning stage, as diversity in features was prioritized, but not so in the post-learning phase.

- ii) Our case is unique in that we needed to combine all the feature rankings from the 28 base-learners into a single ensemble ranking. This necessitated us to have a unified feature ranking method that can be used for different learning algorithms. This provides a consistent metric that then enables us to take the average feature importance values to create a single ranking for the ensemble.
- iii) For the derived personalized risk factors (Fig. 7b), we needed a ranking method that is able to provide local/case-wise feature importance's for each patient and each base-learner. While RF/ET provide case-wise feature importance, XGB is not able to provide such, and even if so, the importance metric across algorithms wouldn't be consistent for combination.

SHAP solves the inconsistency problem mentioned in (i)^{16,17}, provides a single metric for feature importance across different learning algorithms necessitated in (ii) and provides local interpretability meaning we can derive personalized risk factors as required in (iii)^{16,17}. An alternative to using SHAP is LIME¹⁸ as it satisfies both (ii) and (iii), but the consistency property (i) that SHAP guarantees, may be violated in certain instances with LIME¹⁷.

We have re-phrased the **Risk Factors** section to detail our rationale behind the choice of SHAP in the '**Online Methods – Risk Factors**': - Page 20/21

To rank features according to their importance in the model prediction, we used SHapley Additive exPlanations (SHAP). SHAP is a unified framework for explaining the output of any machine learning model^{30,55}. For the post-learning stage, it was important to understand how the features contributed to the predictive performance of CLL-TIM. Therefore, we were limited to using the in-built feature importance of each base-learner in CLL-TIM, such as XGB Feature Selection (FS), RF-FS etc. These methods do not satisfy the feature attribution property of consistency, i.e. a model may rely more than another model on a given feature, but it may still assign a lower importance^{30,78}. Additionally, we needed a method that provided a consistent metric enabling us to generate the average feature importance values to create a single ranking for the ensemble. For the derivation of personalized risk factors, we needed a ranking method that could provide local/case-wise feature importance's for each patient and each base-learner. SHAP satisfies the feature attribution property of consistency^{30,78}, provides a single metric for feature importance across different learning algorithms and provides local interpretability for derivation of personalized risk factors^{30,78}. An alternative to using SHAP is LIME⁷⁹, as it satisfies most of our requirements but the consistency property of feature attribution that SHAP guarantees, may be violated in certain instances with LIME³⁰.

7- For the risk factors identification, the authors should also use some traditional univariate biostatistics techniques, Pearson correlation coefficient, Kruskal-Wallis test, Whitney-Mann U test, Wilcoxon signed-rank test, Student's t-test, chi squared test.

We thank you for this recommendation. We have now added feature rankings using univariate techniques of Student's t-test, ANOVA F-Value, Mann Whitney U-test and Kruskal-Wallis (**Supplementary Tables 4-6**). We were satisfied to see that the risk factors we identified in Fig 6a-b were also confirmed using these univariate techniques. Additionally, from the univariate analysis of the 7288 features developed in this work, infection features were actually the most discriminatory for predicting risk of infection, further

emphasizing the importance of medical and infectious history for CLL-TIM. Pearson correlation coefficient was not performed, as our target output is a binary outcome. Wilcoxon signed-rank test, and chi square test were not performed as the patients in the high-risk and low-risk categories are not paired.

We have updated the manuscript with our findings from the univariate analysis presented in **Supplementary Tables 4-6**:

Results - CLL-TIM Identifies General Risk Factors - Page 8:

This idea of conditional risk contrasts greatly to the independent and additive contributions of factors in prognostic indices such as CLL-IPI, with β 2-microglobulin either contributing two or zero points¹¹. From univariate analysis of the 7288 features generated in this work, we found that for CLL-TIM's top 50 features, 92% were significant ($p < 0.05$) in at least one univariate test, while 72% were significant ($p < 0.05$) in all four univariate tests (Supplementary Table 4). We also observed homogeneity for the features exhibiting the lowest p-values with each univariate test (Supplementary Table 5). These consisted of features based on clinical stage, β 2-microglobulin, hemoglobin and leukocytes. We also performed the same univariate analysis but with infection (as a first event) as the only outcome (Supplementary Table 6). Here, all four univariate tests were dominated by infection related features. Composite outcome risk factors (See Fig. 6a) were confirmed significant in all four univariate tests, except for three that were significant in only two of the four univariate tests. From the 15 infection risk factors (Fig. 6b), 13 were significant in two or more univariate tests (Supplementary Table 6).

8- tSNE is an useful dimensionality reduction visualization technique, but it cannot replace traditional machine learning feature ranking methods. Also, the authors should include the results obtained through UMAP (<https://arxiv.org/abs/1802.03426>).

We thank you for this suggestion. We have now added results using UMAP in **Supplementary Fig. 15**. We would also like to clarify that t-SNE was not employed as a feature-ranking method and was not used to generate global or personalized risk factors. Rather, we used t-SNE on the personalized risk factors generated by SHAP to visualize the feature heterogeneity for the top risk factors for high-risk patients. Re-reading that section of the manuscript, we acknowledge that the wording was unclear.

We have rephrased the **Results** section to further clarify the above and with results for UMAP

Results - CLL-TIM Provides Personalized Risk Factors and Uncovers Multiple Etiologies Towards Risk - Page 9:

To analyze the variability, if any, in the personalized risk factors for patients with a composite outcome, we extracted, from the personalized risk factors of each, the top 3 factors pushing the patients towards high-risk (Fig 7c). Found in 14% of the patients, the most common combination of risk factors was Binet stage, IGHV mutation status and leukocytes maxima in the last 3 months. For 50% of the population, 17 different combinations of 12 risk factors were found in the top 3 personalized risk factors. For characterizing the risk of the remaining 50% of the patients, 313 combinations of 34 risk factors were in turn required – thus highlighting the strength and necessity of having a multiple-outlooks strategy with a heterogeneous set of features. For the 970 patients in the Danish cohort with a composite outcome, we performed clustering of the top 3 personalized risk factors with Stochastic Neighbor Embedding³² (t-SNE – Fig. 7d) and Uniform Manifold Approximation and Projection³³ (UMAP - Supplementary Fig. 15). Results from t-SNE and UMAP

show that patients have distinct clusters of personalized risk factors. Similar to previous comparisons³⁴, UMAP clusters were more defined and less diffused than those with t-SNE.

Other flaws:

9 - The authors say their tool handles efficiently missing data, but they do not explain how.

We thank the reviewer for noticing this important aspect of CLL-TIM that requires further clarification. In summary, we designed CLL-TIM to handle data over multiple levels: the feature level with features that allow missingness to be modelled; the base-learner level with learners that model missingness w/o needing imputation, and on the ensemble level with a design that uses multiple paths to risk and down-weights base-learner predictions with missing data. In further detail:

Feature Level: Baseline variables were one-hot-encoded, hence we have a feature for each baseline variable indicating when it is missing. This allows the learners to model the missing state of a variable just like any other of its states. For the laboratory data, we created features ('Lab Test Date Modeling Variables - See Extended Data Table 1) that are missingness indicators. In this way the learning algorithms, through these features, can take into account recentness of the test dates or the lack there-of (Fig 2b. for the feature encodings of 'Routine Lab Test Date Modelling' selected by CLL-TIM). **Base-Learner Level:** XGBs are capable of creating splits for missing data without imputation¹⁴. For the other methods, we used median imputation for their features. No imputation was performed for baseline variables as these were one-hot-encoded. **Ensemble Level:** The ensemble was designed in such a way to promote diversity in predictions by using several feature selection methods, various learning algorithms trained with randomized learning parameters. Additionally, the ensemble score was designed to rank highly those ensembles that had low correlation in the base-learner predictions. The rationale being that if predictions in the base-learners have low correlation and act on different features, then missing data in a set of features for one patient, may be compensated by available data in another set of features used by different base-learners^{19,20}. Additionally, given that we take the average probabilistic output over all base-learners, this means that we are implicitly down-weighting base-learner predictions with missing data as the probabilistic output of these base-learners would be close to 0.5.

Post-learning, feature diversity was observed in both the ensemble structure (Fig. 2) and the top risk factors for patients (Fig. 7b-d and Supplementary Fig. 15), showing that the compensation mechanism of a diverse ensemble is in place and does use different paths for assigning risk. Additionally, we have now also performed Pearson Correlation Coefficient (PCC) for the 228 features and demonstrated low redundancy among these features, suggesting that CLL-TIM uses complimentary information within features for its predictions (See Supplementary Fig.2). If we opted for a single learning algorithm using a single feature selection method, then patients with no data for the top features would likely give erroneous results or no results at all, as is seen with CLL-IPI when one or more CLL-IPI variables are missing. We have also put significant effort into quantifying and visualizing the missingness of the datasets (Figs. 3f, 3h, Supplementary Fig. 7) and also the benchmarking of CLL-TIM under several missing value simulations (Supplementary Figs. 6-9, Supplementary Fig. 10 and Fig 4). This also includes the external testing of CLL-TIM under high rates of missing data as in the German cohort (Fig. 3e-h).

We have added a new **Online Methods** section entitled 'Handling of Missing Data':

Handling of Missing Data

Missing data was handled over several layers in the design of CLL-TIM. On the feature level, baseline variables were one-hot-encoded, hence we have a feature for each baseline variable indicating when it is missing. For the laboratory data, we created features ('Lab Test Date Modeling Variables - See Extended Data Table 1) that are missingness indicators. In this way the learning algorithms, through these features, can take into account recentness of the test dates or the lack there-of. On the base-learner level, XGBoost is capable of creating splits for missing data without imputation. For the other methods, we used median imputation for their features. No imputation was performed for baseline variables as these were one-hot-encoded. On the ensemble level, the ensemble was designed in such a way to promote diversity in predictions (see 'Methods' - Base-Learner Generation, and Ensemble Ranking), the rationale being that different parts of the ensemble may be able to compensate for missing data in others^{19,20}. Given that we take the average probabilistic output over all base-learners, this means that we are implicitly down-weighting base-learner predictions with missing data.

Results – Development and Composition of CLL-TIM – page 4:

“...from which the top-ranked ensemble, CLL-TIM, was selected as the final model (Supplementary Fig.1). We handled missing data using different methodologies (See Methods 'Handling of Missing Data' (See Methods, 'Handling of Missing Data')). CLL-TIM is composed of 28 base-learners spanning both linear and non-linear algorithms.”

10 - The authors claim they predict "infection or CLL treatment", but this expression is confusing. What does it mean? Is an infection the same thing of a CLL treatment? Are they predicting both? Why are these two concepts treated in the same way?

Thank you for the option to further clarify this important issue. The risk of infection and the risk of CLL treatment are two different but interlinked issues. The “or” in this case means that we are considering a high-risk patient to be one who initiates treatment within the first two years post CLL diagnosis, one who experiences an infection in this time period, or one who experiences both. As highlighted in the introduction, CLL prognostic models have so far focused on the prediction of treatment or overall survival. Yet infection has become the leading cause of mortality in CLL²¹ and hitherto we have had no predictor for risk of infection. So far, we know that patients experiencing an infection within one year of diagnosis have an inferior Overall survival (OS) and Progression Free Survival (PFS)^{7,9}. Furthermore, the risk of infection increases upon treatment with chemoimmunotherapy (unpublished data)²². Thus, the immune dysfunction leading to an increased risk of infection reflects a more aggressive CLL disease linked with higher risk of treatment need (prior to or after and infection). These clinical observations led to the idea of combining infections and CLL treatment into a composite outcome.

Our attempt to understand the link between these two outcomes was originally detailed in Figure 4a, but prompted by this, we have removed Fig. 4a and remodeled this with more clarity in **Figure 5a-c**. Here we assessed whether models trained to predict the treatment outcome are predictive of infection and whether models trained to predict infection are predictive of treatment. We found that modelling infection is predictive of treatment, but modelling treatment was not predictive of infection. The latter is corroborated

by our findings in (Figure 5d, e and Supplementary Fig.14; CLL-IPI, which is designed for prediction of overall survival and time to treatment, is not able to predict risk of infection sufficiently. When we model both outcomes, as performed with the composite outcome, we improve performance for prediction of both treatment and infection (Figure 5a-c). This suggests that a synergy exists when modelling this composite outcome, which is in line with the clinical experience that the two outcomes are interlinked. So even if you want to predict just treatment, or just infection, you are better off by modelling the composite outcome. Furthermore, from a clinical perspective, the event of aggressive and progressive CLL is linked to increased risk of immune dysfunction, thus making the combined outcome clinically relevant.

We have now updated:

Results - Immune Dysfunction is Linked to Progressive Disease – Page 7

To examine the link between infections and CLL treatment as outcomes, we compared ensemble models trained to predict the composite outcome to models that were trained to predict CLL treatment and models predicting infection as a first event (ENS-COMP, which includes the CLL-TIM ensemble; ENS-TREAT and ENS-INFEC, Extended Data Table 3). Training on infection as an outcome in combination with CLL treatment (as in ENS-COMP and CLL-TIM), synergistically improves the predictions of CLL treatment ($p < 0.05$ for ENS-COMP vs. ENS-TREAT, Fig 5a). This result was corroborated by CLL-TIM also outperforming CLL-IPI in predicting CLL treatment on both internal and external cohorts (Supplementary Figs. 11 and 12). Considering next, patients with infection prior to CLL treatment as an outcome, modeling only CLL treatment was not predictive (Recall 7% - Fig. 5b). However, both ENS-INFEC and ENS-COMP models were predictive and had significantly similar recalls of around 40% ($p > 0.05$, Fig 5b). In brief, modeling only CLL treatment was not enough to predict infection prior to CLL treatment, and modeling infection as an outcome, while necessary for the prediction of infections, was also somewhat predictive of CLL treatment (See Supplementary Fig. 13 for further evidence). Similar to ENS-TREAT, CLL-IPI was not predictive of infections prior to CLL treatment (Supplementary Figs. 14 a and b). We next wanted to rule-out whether this was due to CLL-IPI variables not being predictive of infection, or due to them not being trained to predict infection as an outcome. For this, we trained ensembles with CLL-IPI variables to predict both infection and treatment. Similar to CLL-IPI, these ensembles were also not predictive of infection (Supplementary Figs. 14c and d). We next assessed how the recall for infections, differs when changing the definition for this outcome (Fig. 5). For high-confidence predictions, recall increased for patients with positive or multiple blood cultures. However, none of these increases were significant ($p > 0.05$). In turn, for all outcomes, CLL-TIM achieved significantly higher recalls than CLL-IPI.

Discussion - Page 10:

A link between immune dysfunction and CLL aggressiveness has also been established in our data driven approach. Analyses aimed at understanding the link between the risk of infection and risk of treatment showed that modeling infection together with CLL treatment was necessary for the detection of infection prior to CLL treatment but also improved the detection of CLL treatment prior to infection (Fig. 5a-c). Thus, by combining the two clinically interlinked outcomes of immune dysfunction leading to risk of infection and risk of CLL treatment into a joined outcome, we were able to draw mutually predictive information from both event types. This, also improved predictions for both outcomes.

11 - The grammar is good but all the passive verb forms must be replaced with active verb forms.

We thank you for this observation and have rephrased such instances in the manuscript and highlighted them in red within the manuscript.

Bibliography

1. *Andersen, M. A., Moser, C. E., Lundgren, J. & Niemann, C. U. Epidemiology of bloodstream infections in patients with chronic lymphocytic leukemia: a longitudinal nation-wide cohort study. Leukemia 33, 662–670 (2019).*
2. *Fournier, P.-E. et al. Blood culture-negative endocarditis: Improving the diagnostic yield using new diagnostic tools. Medicine 96, e8392 (2017).*
3. *Salimnia, H. et al. Evaluation of the filmarray blood culture identification panel: results of a multicenter controlled trial. J. Clin. Microbiol. 54, 687–698 (2016).*
4. *Iroh Tam, P.-Y. et al. Detection of Streptococcus pneumoniae from culture-negative dried blood spots by real-time PCR in Nigerian children with acute febrile illness. BMC Res. Notes 11, 657 (2018).*
5. *Nordvig, J. et al. Febrile Neutropenia and Long-term Risk of Infection Among Patients Treated With Chemotherapy for Malignant Diseases. Open Forum Infect. Dis. 5, ofy255 (2018).*
6. *Andersen, M. A. et al. Incidence and predictors of infection among patients prior to treatment of chronic lymphocytic leukemia: a Danish nationwide cohort study. Haematologica 103, e300–e303 (2018).*
7. *Crassini, K. R., Best, O. G. & Mulligan, S. P. Immune failure, infection and survival in chronic lymphocytic leukemia. Haematologica 103, e329 (2018).*
8. *Crassini, K. R. et al. Humoral immune failure defined by immunoglobulin class and immunoglobulin G subclass deficiency is associated with shorter treatment-free and overall survival in Chronic Lymphocytic Leukaemia. Br. J. Haematol. 181, 97–101 (2018).*
9. *Andersen, M. A. & Niemann, C. U. Immune failure, infection and survival in chronic lymphocytic leukemia in Denmark. Haematologica 103, e330 (2018).*
10. *de Brébisson, A., Simon, É., Auvolat, A., Vincent, P. & Bengio, Y. Artificial Neural Networks Applied to Taxi Destination Prediction. arXiv (2015).*
11. *Bishop, C. M. Pattern Recognition And Machine Learning (information Science And Statistics). 738 (Springer, 2006).*
12. *Fabian Pedregosa et al. Scikit-learn: Machine Learning in Python %J J. Mach. Learn. Res. 12, 2825–2830 (2011).*
13. *Breiman, L., Friedman, J. H., Olshen, R. A. & Stone, C. J. Classification and regression trees. (Wadsworth & Brooks/Cole Advanced Books & Software, 1984). doi:10.1201/9781315139470*

14. *Chen, T. & Guestrin, C. XGBoost: A Scalable Tree Boosting System. in Proceedings of the 22nd ACM SIGKDD International Conference on Knowledge Discovery and Data Mining - KDD' '16 785–794 (ACM Press, 2016). doi:10.1145/2939672.2939785*
15. *Breiman, L. Random Forests. Springer Science and Business Media LLC (2001). doi:10.1023/a:1010933404324*
16. *Lundberg, S. M., Erion, G. G. & Lee, S.-I. Consistent Individualized Feature Attribution for Tree Ensembles. arXiv (2018).*
17. *Lundberg, S. M. & Lee, S.-I. A Unified Approach to Interpreting Model Predictions. (2017).*
18. *Ribeiro, M. T., Singh, S. & Guestrin, C. Why should I trust you?": Explaining the predictions of any classifier. in Proceedings of the 22nd ACM SIGKDD International Conference on Knowledge Discovery and Data Mining 1135–1144 (ACM Press, 2016). doi:10.1145/2939672.2939778*
19. *Melville, P., Shah, N., Mihalkova, L. & Mooney, R. J. Experiments on ensembles with missing and noisy data. 3077, 293–302 (2004).*
20. *Polikar, R. Bootstrap - Inspired Techniques in Computation Intelligence. IEEE Signal Process. Mag. 24, 59–72 (2007).*
21. *da Cunha-Bang, C. et al. Improved survival for patients diagnosed with chronic lymphocytic leukemia in the era of chemo-immunotherapy: a Danish population-based study of 10455 patients. Blood Cancer J. 6, e499 (2016).*
22. *Eichhorst, B. et al. First-line chemoimmunotherapy with bendamustine and rituximab versus fludarabine, cyclophosphamide, and rituximab in patients with advanced chronic lymphocytic leukaemia (CLL10): an international, open-label, randomised, phase 3, non-inferiority trial. Lancet Oncol. 17, 928–942 (2016).*

REVIEWERS' COMMENTS:

Reviewer #1 (Remarks to the Author):

no comments

Reviewer #2 (Remarks to the Author):

The authors correctly addressed my comments and suggestions and edited the manuscript accordingly.

I have no other request.